# State of the Art of Techno-Economics of Nanofluid-Laden Flat-Plate Solar Collectors for Sustainable Accomplishment

**Seyed Reza Shamshirgaran [1]** , **Hussain H. Al-Kayiem [2]** , **Korada V. Sharma [3,\*]** and **Mostafa Ghasemi [4]**

[1]  Department of Renewable Energy Engineering, Faculty of Mechanical and Energy Engineering, Shahid Beheshti University, Tehran 1983969411, Iran; r_shamshirgaran@sbu.ac.ir

[2]  Mechanical Engineering Department, Universiti Teknologi PETRONAS, Seri Iskandar 32610, Malaysia; hussain_kayiem@utp.edu.my

[3]  Mechanical Engineering Department, JNTUH College of Engineering, Telangana 500085, India

[4]  Chemical Engineering Section, Sohar University, Sohar 311, Oman; MBaboli@su.edu.om

\*  Correspondence: kvsharmaump@gmail.com; Tel.: +91-98490-99162

**Abstract:** Emerging nanotechnology with solar collector technology has attracted the attention of researchers to enhance the performance of solar systems in order to develop efficient solar thermal systems for future sustainability. This paper chronologically reviews the various research works carried out on the performance enhancement of nanofluid-filled flat-plate solar collectors (FPCs). Gaps in the radiation exergy models and maximum exergy of FPCs, the importance of pressure drops in collector manifolds in exergy analysis, and the economics of nanofluid-laden FPCs have been addressed. The necessity of replacing currently used chemically derived glycol products with a renewable-based glycol has not been reported in the current literature thoroughly, but it is pondered in the current paper. Moreover, the thermophysical properties of all common metal and metal oxide nanoparticles utilized in various studies are collected in this paper for the first time and can be referred to quickly as a data source for future studies. The different classical empirical correlations for the estimation of specific heat, density, conductivity, and viscosity of reported nanofluids and base liquids, i.e., water and its mixture with glycols, are also tabulated as a quick reference. Brief insights on different performance criteria and the utilized models of heat transfer, energy efficiency, exergy efficiency, and economic calculation of nanofluid-based FPCs are extracted. Most importantly, a summary of the current progress in the field of nanofluid-charged FPCs is presented appropriately within two tables. The tables contain the status of the main parameters in different research works. Finally, gaps in the literature are addressed and mitigation approaches are suggested for the future sustainability of nanofluid-laden FPCs.

**Keywords:** exergy efficiency; nanofluid; nano-thermophysical properties; sustainability of solar collectors; techno-economic

---

## 1. Introduction

Flat-plate solar collectors (FPCs) are regarded as a good alternative to meet the thermal energy demand from the domestic and industrial sectors. Thermal performance enhancement of solar FPCs is progressing either by advancing the absorber plate or improving the heat transfer fluid (HTF) effectiveness. Both methods deal with energy and exergy efficiency of the collector. Possessing a low exergy efficiency has made FPCs notorious in that their work extraction capability is not significant compared to other solar collectors. Regarding the upper limit for conversion of unconcentrated solar

radiation, which is 5.3% [1], the exergy efficiency of an FPC might not be higher than 5% [2–4]. The challenge of low exergy efficiency has continued from the past from theoretical [3,4] and experimental [5–7] points of view. That is why enormous investigations seeking appropriate solutions to boost both energetic efficiency and exergetic efficiency of FPCs have yet to be published. Blackbody absorbers are not recommended for efficiency enhancement, since their absorptance and emittance are equal. Therefore, spectrally selective absorbers, with an absorptance higher than their emittance, are considered as an effective method to improve the efficiency of FPCs. Presently, absorbers with highly selective coatings are made from copper and aluminum, which provide the absorptivity and emissivity of 95% and 4%, respectively, at 100 °C [8]. The possibility of utilizing different materials to provide a high selectivity at higher temperatures is also under investigation [2,9–14].

As the second method of efficiency improvement, the addition of nanomaterials to the HTF is a result of relatively recent progress. This approach enhances the conductivity of the collector's HTF and boosts the heat transfer coefficient and thermal efficiency. However, the viscosity will also increase, and this, in turn, may affect the exergy efficiency, pumping power, and economics of the collector operation [15]. The Carnot and Petela–Landsberg–Press (PLP) models are well-known equations that have long been used to calculate the exergy of solar radiation. However, their application is rigorously restricted to cases where the conversion of radiation energy into work is performed by using blackbody collectors, not selective absorbers. The necessity of sustainability has propelled investigators to substitute environmentally harmful products with green ones. The current glycol products, namely ethylene glycol (EG) and propylene glycol (PG), which are used as antifreeze or stabilizer base fluids in FPCs, are chemically derived and might even be toxic. The application of a renewable-based alternative is essential.

Previous reviews [16–29] have not comprehensively covered the exploited radiation exergy model, the value of absorptance and emittance of the utilized absorber, the base liquid type, the importance of pressure drops in collector manifolds, and, consequently, the exergy destruction due to the pressure drop. It is also realized that the detailed economics of application of nanomaterials in FPCs have not been comprehensively reported and compared in one single compilation. Moreover, the flow regime and change in overall efficiency of the collector have not been reported clearly in many research studies. That is why the present review paper addresses all the aforementioned points to provide clear insights into the status of the many studies that have reported the enhancement of nanofluid-based FPCs. Furthermore, there is not a data source that has gathered the thermophysical properties of all common nanoparticles used in different studies. So far, there is no comprehensive collection of the various models for estimation of thermophysical properties of nanofluids and heat transfer calculations. The current review paper deals with all of this subject matter to provide a complete reference as a summary of investigations carried out in the field of nanofluid-based solar FPCs. Future research directions are also drawn from the existing gaps.

## 2. Solar Thermal Collectors for Water Heating

Solar thermal collectors convert the radiation coming from the sun into thermal energy for heating an HTF, which is an effective solution for conservation of energy. The temperature level and the amount of this conversion depend on the collector technology [30]. Generally, solar thermal collectors are classified into non-concentrating and concentrating types according to the degree of concentration. Solar thermal collectors are classified as shown in Figure 1.

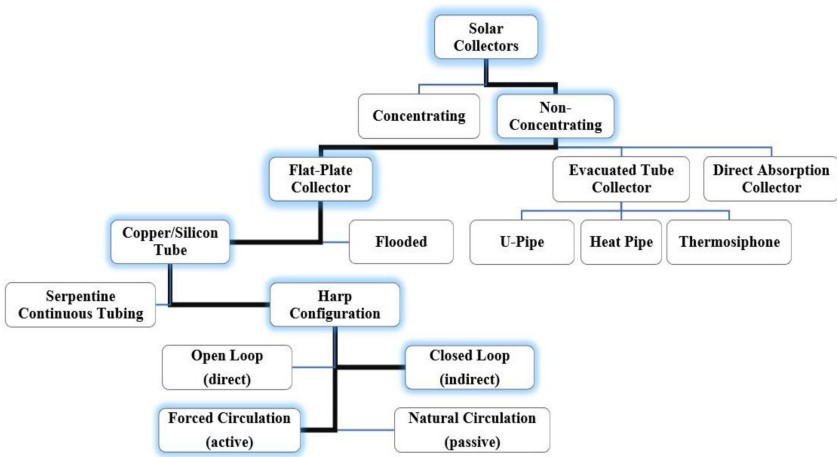

**Figure 1.** Classification of solar thermal collectors.

Concentrating collector technology is categorized as medium concentration, such as with a parabolic cylinder, and high concentration, such as with paraboloidal collectors. A concentrating collector utilizes a reflective parabolic-shaped surface to reflect and concentrate the solar energy to a focal point or focal line where the absorber is located. A concentrating collector can be designed as a fixed device or with rotating reflectors to be able to track the sun's position in the sky for maximum solar capture and effective working. These collectors can achieve very high temperatures because the diffuse solar resource is concentrated in a small area. The temperature levels that can be provided by different solar collectors are given in Table 1.

**Table 1.** Possible temperature level based on concentration degree of solar collectors.

| Category | Example | Temperature Range (°C) | Efficiency (%) |
|----------|---------|------------------------|----------------|
| Non-concentrating | Flat-plate | up to 75 | 30–50 |
| | Evacuated tube | up to 200 | |
| Medium concentrating | Parabolic cylinder | 150 to 500 | 50–70 |
| High concentrating | Paraboloidal | 1500 and more | 60–75 |

An evacuated tube collector (ETC) consists of a number of glass tubes. During the manufacturing process, a vacuum is created inside the glass tubes to reduce heat loss through conduction and convection. Since two flat sheets of glass are normally not strong enough to withstand a vacuum, the vacuum is rather created between two concentric tubes. U-pipe ETCs are equipped with two pipes that run down and back, inside the tube. One pipe is for inlet fluid and the other for outlet fluid. Since the fluid flows into and out of each tube, the tubes are not easily replaced.

Heat pipe ETCs contain a copper heat pipe, which is attached to an absorber plate, inside a vacuum sealed solar tube. The heat pipe is hollow, and the space inside is also evacuated. Inside the heat pipe is a small quantity of liquid, such as alcohol or purified water, plus special additives. The vacuum enables the liquid to boil at lower temperatures than it would at normal atmospheric pressure. When sunlight falls on the surface of the absorber, the liquid in the heat tube quickly turns to hot vapor and rises to the top of the pipe. Water or glycol flows through a manifold and picks up the heat. The fluid in the heat pipe condenses and flows back down the tube. This process continues, as long as the sun shines.

A non-concentrating direct absorption collector (DAC) absorbs all or a significant fraction of solar radiation directly. Usually, there is not a containment surface separating the working fluid from the solar flux [31].

FPCs are the most widely used type of collectors in the world for domestic solar water heating or even solar space heating applications [30]. They use both beam and diffuse solar radiation,

do not require tracking of the sun, and require little maintenance. FPCs are durable and effective. These collectors have a distinct advantage over other types in that they shed snow and rain very well when installed in climates that experience significant snowfall and rainfall. FPCs are divided into tube-type and flooded-type. In flooded-type FPCs, the absorber is designed as two sheets of metal, which allows the HTF to flow between them. The surface area is increased, and the efficiency can be boosted. The tube-type FPCs are of two configurations, namely harp and serpentine. Serpentine FPCs have a continuous s-shaped absorber in which the HTF enters from one side and exits to the other side upon being heated.

The harp-configuration or parallel-scheme of FPCs is the most common design and is composed of header pipes with riser tubes between them. Depending on how the HTF contacts tap water, FPCs are divided into direct and indirect types. The HTF may exchange the heat with tap water directly in a storage tank (open-loop design) or indirectly through coil exchangers in the tank (closed-loop design). The path in which the HTF flows is called the primary or solar circuit. The tap water which absorbs heat from the HTF flows in the secondary circuit.

FPCs are also categorized based on the driving force required for circulation of HTF in the collector. A passive vs. an active FPC refers to whether the HTF is circulated by pumps (active) or by thermodynamics (passive). Passive systems are most common in regions where temperatures below freezing are not common. In these systems, water naturally circulates by convection, known as thermosiphoning, eliminating the need for a pump. Active designs use pumps to move the fluid through the collector.

## 3. Energy Conversion in a Solar FPC

Anything that is exposed to solar radiation can be called a solar collector. A solar FPC is made up of many parts; however, the main components of an FPC are a cover, combined absorber and riser, and insulation, as shown in Figure 2.

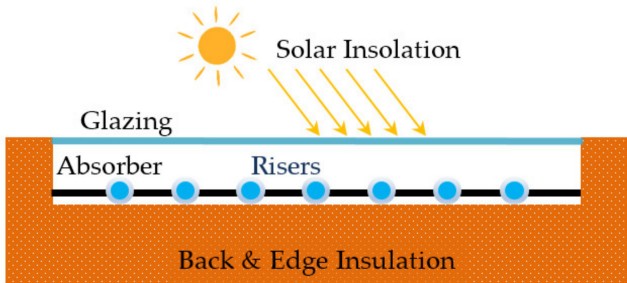

**Figure 2.** Schematic diagram of a flat-plate solar collector (FPC) structure.

The principal energy gain and loss mechanism for a solar FPC applied to all solar thermal collectors is shown in Figure 3. The task of a solar collector is to convert solar radiation with an intensity of $G_{tot}$ into useful thermal energy, $\dot{Q}_u$. Part of solar radiation, $G_{loss}$, is lost due to optical losses in collector glazing, and another part is wasted through thermal losses, $\dot{Q}_{loss}$. Of the absorbed heat, net useful thermal power, $\dot{Q}_{out}$, is delivered for a specific application at a suitable temperature and with certain efficiency.

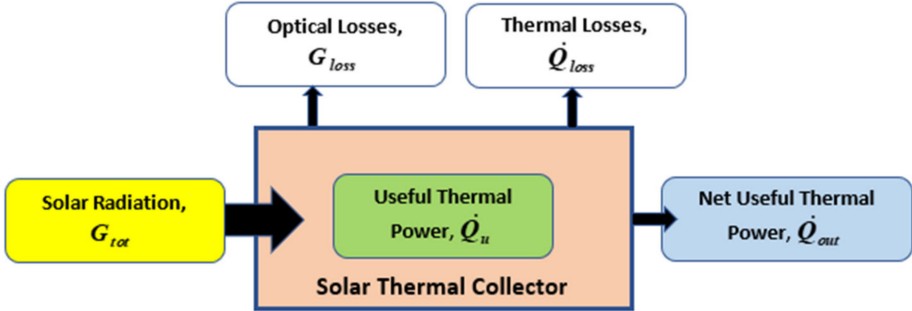

**Figure 3.** Principle of energy flow in a solar flat-plate collector.

Therefore, two main efficiencies, namely optical and thermal, can be considered for the energy conversion in solar thermal collectors, as shown in Figure 4.

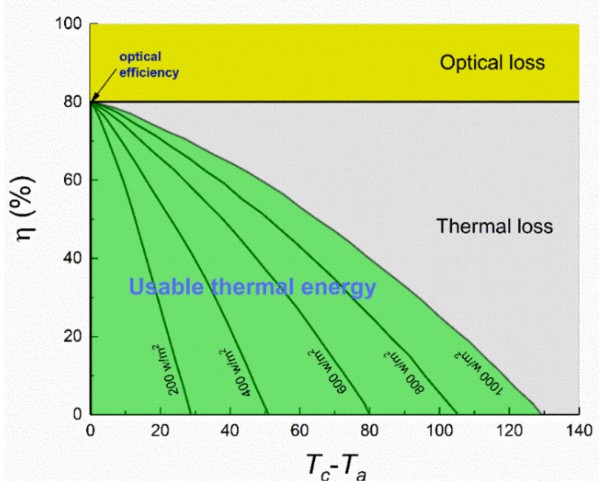

**Figure 4.** Efficiency of solar collectors at various temperature differences and solar insolation [32].

The optical efficiency of a solar FPC is dependent on transmittance of its glazing cover and absorptance of its absorber plate. Thermal efficiency is described based on the fraction of received solar energy, which is delivered as usable thermal energy. The curves of Figure 4 show that thermal losses go up with the increase in temperature difference between the collector and ambient air. Meanwhile, the rate of efficiency decrease in low solar irradiances is higher. At the point at which the collector temperature and ambient temperature are the same, there is no heat loss. The efficiency at this point is called zero-loss efficiency, and it equals the optical efficiency of the collector. Therefore, the performance of an FPC at this point exclusively depends on its optical behavior.

In the example of the solar collector shown in Figure 4, the optical efficiency is around 80%. Heat transfer enhancement by the employment of a novel working fluid rather than water assists in diminishing the thermal loss contribution and improving the thermal efficiency of an FPC. The novel working fluid, i.e., nanofluid, is realized through the conjunction of nanotechnology with solar collector technology [33].

## 4. Nanotechnology and Nanomaterials for FPCs

Currently, there are two definitions for nanotechnology. The first one [34] states that nanotechnology is the comprehension and control of matter at dimensions between approximately 1 and 100 nm, where unique phenomena empower novel applications. Encompassing nanoscale science, engineering, and technology, nanotechnology involves imaging, measuring, modelling, and manipulating matter at this length scale. The dimensional range between 1 and 100 nm is known

as the nanoscale. Uncommon physical, chemical, and biological properties can arise in materials at the nanoscale. These properties may contrast from the properties of bulk materials and single atoms or molecules. The second definition expresses that nanotechnology is the design, characterization, production, and application of structures, devices, and systems through the controlled manipulation of size and shape at the nanoscale (atomic, molecular, and macromolecular scale) that produce structures, devices, and systems with at least one novel/superior characteristic or property [34]. Therefore, it comes to mind that the sub-100 nm size constraint of nanotechnology cannot accept all existing materials and devices. Furthermore, definition necessitates that the nanostructure is synthetically produced.

Despite various definitions, nanomaterials are known as materials that are at the nanoscale in at least at one dimension [35,36]. One-dimensional (1D) nanoscale materials are called layers, e.g., graphene, thin films, or surface coatings. Two-dimensional (2D) nanoscale materials are called nanowires and nanotubes. The materials that are nanoscale in three dimensions are considered as particles, colloids, and quantum dots (tiny particles of semiconductors). Nanocrystalline materials, made up of nanoscale grains, also fall into this category. The properties of nanomaterials differ remarkably from other materials due to two key factors: enhanced relative surface area, and quantum effects. The factors can alter or enhance characteristics such as reactivity, strength, and electrical properties. When decreasing a particle size, a greater number of atoms exist at the particle surface compared to those inside the particle. For instance, a 30-nm size particle that has 5% of its atoms on its surface may have 20% and 50% of its surface atoms at a size of 10 nm and 3 nm, respectively [34]. Hence, nanomaterials have a much greater surface area per unit mass compared to larger particles. In other words, when growth and catalytic chemical reactions happen at surfaces, a certain mass of material in the nanoscale will be much more reactive than the same mass of material comprised of larger particles. To gain a better understanding, imagine a silver coin containing 31 g of silver and having a total surface area of around 3000 $mm^2$. If the same amount of silver were divided into nanoparticles with diameters of 10 nm, then the total surface area of those particles would be 7000 $m^2$. This value is equal to the size of a football field. In other words, when the silver content of a silver dollar is changed to 10 nm particles, the particles' total surface area is over 2 million times greater than that of the silver dollar.

Together with the effect of surface-area, the quantum effect can dominate the properties of matter as size is reduced to the nanoscale. These effects can influence the optical, electrical, and magnetic behavior of materials, particularly as the structure or particle size approaches the smaller end of the nanoscale. For other materials such as crystalline solids, as the size of their structural components decreases, there is much greater interface area within the material; this can greatly affect both mechanical and electrical properties. The thermophysical properties of different nanomaterials are given in Table 2. The data, including thermal conductivity, heat capacity, and density, are adapted from different resources [37–71]. It should be noted that some of the values presented in Table 2 may differ slightly in other references. This is due to the different temperatures at which the properties were measured and reported. However, it is common to report the properties of nanomaterials at standard room temperature.

**Table 2.** Thermophysical properties of different nanomaterials collected from different sources [37–71].

| Name | Material | $k$ (W/m·K) | $C_p$ (J/kg·K) | $\rho$ (kg/m$^3$) |
|---|---|---|---|---|
| Gold | Au | 317 | 129 | 19,300 |
| Silver | Ag | 430 | 235 | 10,490 |
| Copper | Cu | 400 | 385 | 8933 |
| Nickle | Ni | 90.7 | 444 | 8900 |
| Iron | $\alpha$-Fe | 80.2 | 447 | 7870 |
| Zinc | Zn | 116 | 388 | 7135 |

**Table 2.** *Cont.*

| Name | Material | k (W/m·K) | $C_p$ (J/kg·K) | $\rho$ (kg/m³) |
|---|---|---|---|---|
| Diamond | C | 3300 | 509 | 3530 |
| Aluminum | Al | 237 | 904 | 2700 |
| Silicon | Si | 148 | 714 | 2320 |
| Graphite | C | 120 | 701 | 2160 |
| Sodium | Na | 72.3 | 1230 | 968 |
| Cerium oxide | $CeO_2$ | 6 | 616.4 | 7216 |
| Zinc oxide | ZnO | 29 | 514 | 5600 |
| Tin oxide | $SnO_2$ | 31.38 | 343 | 5560 |
| Zirconia | $ZrO_2$ | 1.7 | 504 | 5500 |
| Hematite iron oxide | $\alpha$-$Fe_2O_3$ | 12.55 | 650.64 | 5260 |
| | $Fe_3O_4$ | 6 | 670 | 5180 |
| Magemite iron oxide | $\gamma$-$Fe_2O_3$ | 5 | 653 | 4870 |
| Ferri hydride (goethite) | $\alpha$-FeOOH | - | - | 4260 |
| Titania | $TiO_2$ | 8.9 | 686 | 4250 |
| Alumina | $\gamma$-$Al_2O_3$ | 40 | 765 | 3970 |
| Manganese oxide | MgO | 55 | 874 | 3580 |
| Silica | $SiO_2$ | 1.4 | 745 | 2220 |
| Hybrid | MgO + Ag | 242 | 554.5 | 7035 |
| Hybrid | MWCNT + $Fe_3O_4$ | 509.14 | 680.66 | 4845.4 |
| Silicon carbide | SiC | 490 | 675 | 3160 |
| Titanium carbide | TiC | 330 | 711 | 4930 |
| Carbon nanotube | CNT | 3000 | | 1350 |
| Single-wall CNT | SWCNT | 3500 | 1380 | 1400 |
| Multi-wall CNT | MWCNT | 15 | 470 | 2100 |
| Aluminum nitride | AlN | 285 | 740 | 3260 |

### 4.1. Concept of Nanofluids

It has long seemed a reasonable idea to add particles to a base fluid to enhance thermal conductivity [72]. However, in all early investigations, suspensions of millimeter- or micrometer-sized particles were used by researchers. These led to problems such as poor suspension stability and hence channel clogging, often creating serious problems for systems with small channels. Therefore, special attention was given to nanometer sizes, and in this way, nanofluids with sizes less than 100 nm as new promising HTFs were invented. The efforts are still ongoing to promote the concept and applications, so that, for instance, recently it has been found that Chinese ink nanofluid is a high-performance, cheap, and simply-prepared fluid for solar applications [73], similar to Indian ink proposed earlier in 1975 [74]. Nanofluid as a novel heat transfer medium has attracted the attention of many researchers around the world.

### 4.2. Definition of Nanofluid

Basically, nanofluid is defined as a colloidal suspension of nanometer-sized materials, i.e., nanoparticles or nanosized particles with the size of 1 to 100 nm, nanofibers, nanotubes, nanowires, nanorods, nanosheets, or droplets in liquids. In other words, nanofluids, which considered to

be two-phase systems, are nanoscale colloidal suspensions containing condensed nanomaterials. Nanofluid is composed of three components, viz. base (mother) liquid, nanoparticle, and surfactant.

*4.3. Classification of Nanofluid*

Different types of these components may be combined together to form a new nanofluid, as shown in Figure 5 [75,76]. Common base liquids are water, ethylene glycol, propylene glycol, engine (mineral) oil, and acetone. Nanomaterials are divided into three categories as follows:

(1) Organic nanomaterials (fullerene, nanotube, electrospun),
(2) Inorganic nanomaterials (metal, metal oxides, ceramics, quantum dots), and
(3) Hybrid nanomaterials.

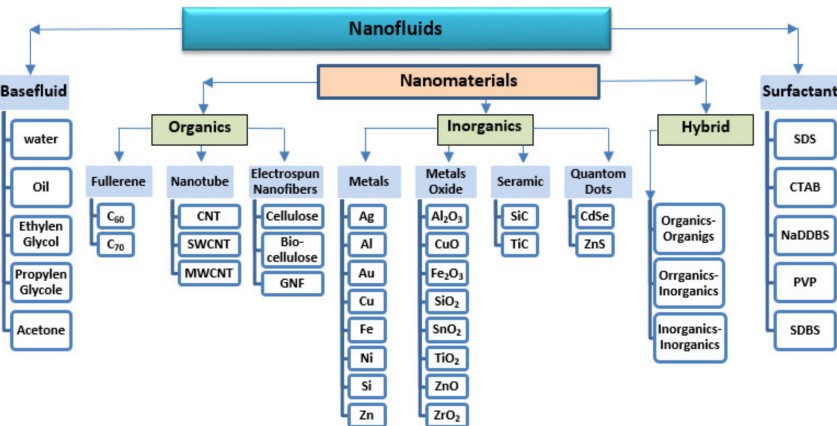

**Figure 5.** Surfactant additives are utilized to increase the stability of nanofluids and to improve their dispersion behavior (SDS: sodium dodecyl sulphate, CTAB: cetyl trimethyl ammonium bromide, NaDDBS: sodium dodecyl benzene sulfonate PVP: polyvinyl pyrrolidone, SDBS: sodium dodecyl benzene sulfonate).

*4.4. Preparation of Nanofluid*

Nanofluids can be prepared in two ways, namely two-step and one-step methods [65,77]. In the two-step method, nanomaterials are produced as dry powder by chemical/physical methods. Nanosized powder is dispersed into a fluid with the help of intensive magnetic force agitation, ultrasonic agitation, high-shear mixing, homogenizing, and ball milling. An important technique to enhance the stability is the use of surfactants for temperatures less than 60 °C. This is the most economic method, especially for large scale preparation [65].

A one-step method is defined as the simultaneous making and dispersing of the particles in the fluid. In this method, the processes of drying, storage, transportation, and dispersion of nanoparticles are avoided, and thus the agglomeration of nanoparticles is minimized, and the stability of fluids is increased. The method can be used to prepare uniformly-dispersed nanoparticles, and the particles can be suspended in the base fluid in a stable form. The main drawback of one-step method is the impurity effect. The residual reactants are left in the nanofluids due to incomplete reaction or stabilization. The vacuum-SANSS (submerged arc nanoparticle synthesis system) is another efficient method to prepare nanofluids using different dielectric liquids. Different morphologies are determined by various thermal conductivity properties of the dielectric liquids [65]. The prepared nanoparticles exhibit needle-like, polygonal, square, and circular morphological shapes. This method avoids undesired particle aggregation rather well.

### 4.5. Concentration of Nanofluid

There are different ways to represent the concentration of nanofluids. One is the volume fraction, $\phi$, which is defined as the ratio of the volume of a constituent, $V_i$, to the volume of all constituents of the mixture, $V_j$, before mixing [78].

$$\phi_i = \frac{V_i}{\sum_j V_j} \tag{1}$$

The concept is the same with volume percent (vol.%), but the former is expressed as only a number, without a unit, and the latter has a percentage unit. The volume fraction is equivalent to the volume concentration if the solution is ideal, i.e., the volumes of the constituents are additive (the volume of the solution is equal to the sum of the volumes of its ingredients). The sum of all volume fractions of a mixture is equal to 1, as shown in the following equation:

$$\sum_{i=1}^{N} V_i = V; \sum_{i=1}^{N} \phi_i = 1 \tag{2}$$

Volume percent is usually applied for cases in which the solution is a mixture of two fluids. However, percentages are only additive for ideal gases [79]. Sometimes, in case of ethanol and water for instance, which are miscible in all proportions, the designation of solvent and solute is arbitrary. The volume of such a mixture is slightly less than the sum of the volumes of the components. Thus, using the definition above, the term "40% alcohol by volume" refers to a mixture of 40 volume units of ethanol with enough water to make a final volume of 100 units, rather than a mixture of 40 units of ethanol with 60 units of water.

The mass fraction, $\varphi_i$, or percentage by weight, wt.%, is another way of expressing the composition of a mixture with a dimensionless quantity. Mass fraction is defined as follows:

$$\varphi_i = \frac{m_i}{\sum_j m_j} \tag{3}$$

Taking the relation of volume, mass, and density into account, the volume fraction for nanofluid can be converted to its mass fraction and vice versa using the factor $m_{bf}$.

$$\frac{m_{np}/\rho_{np}}{\left(m_{np}/\rho_{np}\right) + \left(m_{bf}/\rho_{bf}\right)} = \frac{\left(m_{np}/m_{bf}\right)/\rho_{np}}{\left(m_{np}/m_{bf}\right)/\rho_{np}) + \left(m_{bf}/m_{bf}\right)/\rho_{bf})} \tag{4}$$

$$\varphi = \frac{\phi/\rho_{np}}{\phi/\rho_{np} + 1/\rho_{bf}} [vol.\%] \tag{5}$$

$$\phi = \frac{\varphi}{1-\varphi} \frac{\rho_{np}}{\rho_{bf}} [wt.\%] \tag{6}$$

Here, the subscripts *np* and *bf* stand for nanoparticle and base fluid, respectively. In order to achieve a nanofluid's specific volume fraction, the required mass of nanoparticles is calculated for a certain volume of base liquid, 100 mL for example, as follows [80]:

$$\frac{V_{np}}{\left(V_{np} + V_{bf}\right)} = \frac{m_{np}/\rho_{np}}{\left(m_{np}/\rho_{np}\right) + 100)} \tag{7}$$

Then, the required mass of nanoparticle $m_{np}$ at each certain value of $\phi$ is obtained.

### 4.6. Thermophysical Properties of Nanofluids

Nanofluid was first conceptualized by Maxwell [81] in 1881. However, the prevailing technical constrains at that time did not allow researchers to validate Maxwell's theory until Choi, in 1995 [82], undertook experiments to determine the properties of nanofluids named by him for the first time [83]. The properties of base liquid, nanomaterial, and surfactant make the thermophysical properties of a nanofluid. The main thermophysical properties of a nanofluid include density, $\rho$ (kg/m$^3$), specific heat capacity, $C_p$ (J/kg·K), thermal conductivity $k$, (W/m·K), and dynamic viscosity, $\mu$ (kg/m·s). There have been proposed various models for the evaluation of these properties. The theoretical models for the estimation of density and heat capacity of nanofluids are in good agreement with the experimental data [84]. Therefore, in practice, the theoretical models are used assuredly. However, there are deviations, sometimes more than 25%, between the conductivity and viscosity models for nanofluids and experimental results. Therefore, developed models for the evaluation of conductivity and viscosity are yet to be enhanced. Different affective factors on thermophysical properties, which should be noted, are summarized in diagrams of Figure 6 [83]. Particle size, shape, and material; and nanofluid volume fraction, pH, and temperature are the most important factors in the modelling of nanofluids.

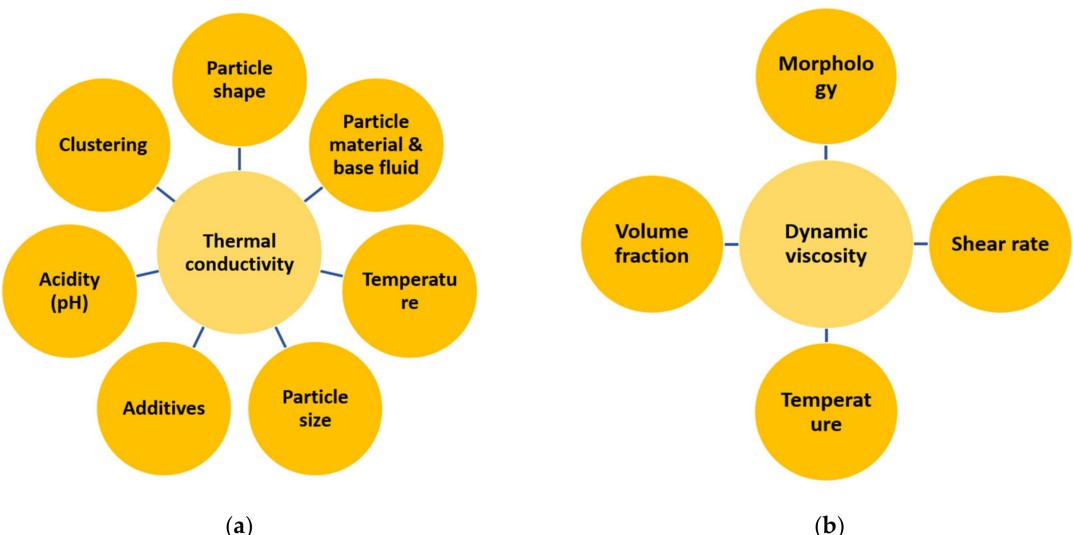

(**a**)          (**b**)

**Figure 6.** Factors affecting conductivity (**a**) and viscosity (**b**) of nanofluids.

### 4.6.1. Density Models of Nanofluids

Generally, the density of a nanofluid can be estimated based on the principle of mixtures' rule as follows:

$$\rho_{nf} = \phi\rho_{np} + (1-\phi)\rho_{bf} \tag{8}$$

However, for some certain nanofluids, there are empirical correlations (see [85,86]).

### 4.6.2. Heat Capacity models of Nanofluids

There are three models for the evaluation of specific heat capacities of nanofluids [87], as given in Table 3.

**Table 3.** Different models for specific heat capacities of nanofluids [87].

| No. | Basis | Model | Validity |
|---|---|---|---|
| Model I | Concepts of mixing theory for ideal gas mixtures | $C_{p,nf} = \phi C_{p,np} + (1 - \phi)C_{p,bf}$ | dilute suspension |
| Mode II | Assumption of thermal equilibrium between nanoparticles and the surrounding base fluid | $\rho_{nf}C_{p,nf} = \phi(\rho_{np}C_{p,np}) + (1 - \phi)(\rho_{bf}C_{p,bf})$ | more accurate |
| Model III | Isobaric specific heat capacity | $C_{p,nf} = \varphi C_{p,np} + (1 - \varphi)C_{p,bf}$ | same as Model I |

Model I is approximately correct only for dilute suspensions for which density differences between the nanofluid and base fluid are small. Model II is more correct and is better fitted with experimental results. Model III is the same as Model I except in terms of concentration basis. Some empirical models can be found for alumina, silica, titania, copper oxide, and zinc oxide nanofluids in [88–91].

### 4.6.3. Thermal Conductivity Models of Nanofluids

The conductivity of a nanofluid can be enhanced by two mechanisms [83]. The first key mechanism is Brownian motion (Figure 7a) and the second is the interfacial nanolayer structure, which is formed nearby the surface of a solid particle (Figure 7b,c).

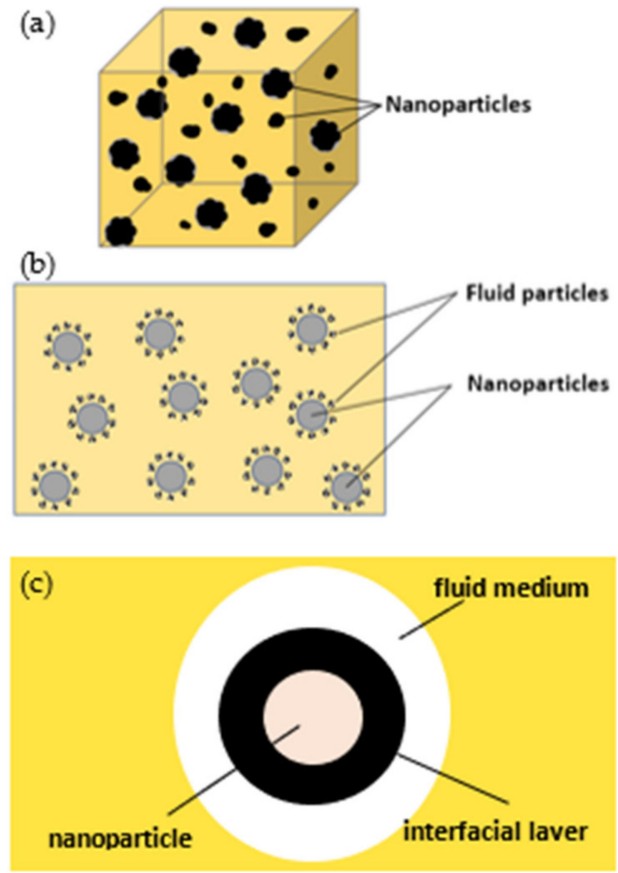

**Figure 7.** Thermal conductivity-enhanced mechanisms of nanofluid nanolayer structure. (**a**) Brownian motion; (**b**) nanofluid structure; and (**c**) nanolayer structure.

Particles, emulsions, and molecules in suspension undergo Brownian motion. This is the motion induced by the bombardment by solvent molecules that themselves are moving due to their thermal energy. The enhancement in the effective thermal conductivity of nanofluids is due mainly to the localized convection caused by the Brownian movement of the nanoparticles. This effect is additive to the thermal conductivity of a static dilute suspension: $K_{eff} = k_{static} + k_{brownian}$ [92]. Since the speed of thermal wave propagation is much faster than the particle Brownian motion, the static part cannot be neglected.

Although liquid molecules close to a solid surface are known to form layered structures, little is known about the connection between this nanolayer and the thermal properties of solid/liquid suspensions. It is assumed that the solid-like nanolayer acts as a thermal bridge between a solid nanoparticle and a bulk liquid and thus is key to enhance thermal conductivity. The main models developed for the conductivity of nanofluids are given in Table 4 [40,83,93–96].

**Table 4.** Different models for conductivity of nanofluids [40,83,93–96].

| Owner | Model | Validity |
|---|---|---|
| Maxwell | $\dfrac{k_{nf}}{k_{bf}} = \dfrac{k_{np}+2k_{bf}-2\phi\left(k_{bf}-k_{np}\right)}{k_{np}+2k_{bf}+\phi\left(k_{bf}-k_{np}\right)}$ | • Theory <br> • Spherical <br> • $\varphi_P \leq 1.0$ |
| Xuan | $\dfrac{k_{nf}}{k_{bf}} = \dfrac{k_{np}+2k_{bf}-2\phi\left(k_{bf}-k_{np}\right)}{k_{np}+2k_{bf}+\phi\left(k_{bf}-k_{np}\right)} + \dfrac{\rho_{np}\phi C_{p,np}}{2k_{bf}}\sqrt{\dfrac{2\kappa_B T}{3\pi\, d_{np} u_{bf}}}$ | • Theory <br> • Spherical and non-spherical <br> • NA |
| Hamilton–Crosser | $k_{nf}/k_{bf} = \dfrac{k_p+(n-1)k_{bf}-(n-1)\phi\left(k_{bf}-k_{np}\right)}{k_{np}+(n-1)k_{bf}+\phi\left(k_{bf}-k_{np}\right)}, n = 3/\psi$ <br> $\psi =$ shape factor <br> $\psi = 1.0$ for spherical nanoparticles <br> $\psi = 0.5$ for cylindrical nanoparticles | • Theory <br> • Spherical <br> • $\varphi_P \leq 4.0$ |
| Wasp | $k_{nf} = \left[\dfrac{k_{np}+2k_{bf}-2\phi\left(k_{bf}-k_{np}\right)}{k_{np}+2k_{bf}+\phi\left(k_{bf}-k_{np}\right)}\right]k_{bf}$ | • Theory <br> • NA <br> • NA |
| Brueggemann | $k_{nf}/k_{bf} = \left[(3\phi-1)\dfrac{k_{np}}{k_{bf}} + (3(1-\phi)-1) + \sqrt{\Delta}\right]/4$ <br> $\Delta = \left[(3\phi-1)\dfrac{k_{np}}{k_{bf}} + (3(1-\phi)-1)\right]^2 + 8\dfrac{k_{np}}{k_{bf}}$ | • Theory <br> • Spherical <br> • NA |
| Yu and Choi | $k_{nf}/k_{bf} = \dfrac{k_{np}+2k_{bf}-2\left(k_{bf}-k_{np}\right)\phi(1+\beta)^3}{k_{np}+2k_{bf}+\left(k_{bf}-k_{np}\right)\phi(1+\beta)^3}$ <br> $\beta =$ ratio of nanolayer thickness to nanoparticle raduis $= 0.1$ | • Theory <br> • Spherical <br> • NA |
| Pack and Cho | $k_{nf}/k_{bf} = 1 + 7.47\phi$ | • Empirical <br> • Spherical <br> • NA |
| Lu and Lin | $k_{nf} = \left(1 + a\phi + b\phi^2\right)k_{bf}$ | • Theory <br> • Spherical and non-spherical <br> • NA |

Among the above classical models, Xuan's equation has shown rather good agreement with empirical data and is frequently used in many investigations. Various empirical models developed for the conductivity of different nanofluids are concluded in reference [40]. All the constraints, especially particle size, material, and concentration, should be taken into consideration prior to using these models.

Among these correlations are the ones proposed by Corcione [96] and Sharma [68], as follows, covering a wide range of applications:

Corcione model : $k_{nf} = [1 + 4.4(Re^{0.4} Pr^{0.66}(\frac{T}{T_{fr}})^{10}(\frac{k_{np}}{k_{bf}})^{0.03}\phi_{np}^{0.66}]k_{bf}; Re = \frac{2\rho_{bf}\kappa_B T}{\pi u_{bf}^2 d_{np}}$

best validity for water-based and EG-based $Al_2O_3$, CuO, Cu, $TiO_2$ nanofluids,
$10 < d_{np} < 150$; $0 < \phi_{np} < 9$; $20 < T < 50$,
$T_{fr} = $ base fluidfreezing point, $\kappa_B = 1.38066 \times 10^{-23}$ J K$^{-1}$

Sharma model : $k_{nf}/k_{bf} = [0.8938(1 + \frac{\phi}{100})^{1.37}(1 + \frac{T_{nf}}{70})^{0.2777}(1 + \frac{d_{np}}{150})^{-0.0336}(\frac{\alpha_{np}}{\alpha_{bf}})^{0.01737}$

best validity for $Al_2O_3$, CuO, $TiO_2$, $Fe_3O_4$, $ZrO_2$, ZnO, and SiC nanofluids,
$20 < d_{np} < 150$; $0 < \phi_{np} < 4$; $20 < T < 70$, $\alpha = $ thermal diffusivity $(m^2/s)$

### 4.6.4. Viscosity Models of Nanofluids

The viscosity of a nanofluid is very important to evaluate because it affects the pressure drop and pumping power. The type of base liquid and nanofluid volume fraction influence the viscosity compared to other factors. The existing classical models for the viscosity estimation of nanofluids are presented in Table 5.

**Table 5.** Existing models for viscosity of nanofluids [40].

| Owner | Model | Validity |
|---|---|---|
| Einstein | $\mu_{nf} = (1 + 2.5\phi)\mu_{bf}$ | • Spherical <br> • $\varphi_P \le 2.0$ |
| Brinkman | $\mu_{nf} = [\frac{1}{(1-\phi)^{2.5}}]\mu_{bf}$ | • Spherical <br> • $\varphi_P \le 5.0$ |
| Batchelor | $\mu_{nf} = (1 + 2.5\phi + 6.2\phi^2)\mu_{bf}$ | • Spherical <br> • NA |
| Graham | $\mu_{nf} = (1 + 2.5\phi + 4.5)[\frac{1}{(\delta/d_p)(2+\delta/d_p)(1+\delta/d_p)^2}]\mu_{bf}$ <br> $\delta = $ interparticle spacing $= \sqrt[3]{\frac{\pi}{6\phi}}d_p$ | • Spherical <br> • NA |
| Kreiger–Dougherty | $\mu_{nf} = [(1 - \frac{\phi_p}{\phi_{p.max}})^{-[\eta]\phi_{p.max}}]\mu_{bf}$ <br> $\eta = 2.5$ for spherical particles <br> $\phi_{p.max} = $ maximum volume fraction $= 0.605 (0.001 \le \phi_P \le 0.05)$ <br> $-[\eta]\phi_{p.max} = -1.5125$ | • Spherical and non-spherical <br> • $0.001 \le \phi_P \le 0.05$ |
| Nielson | $\mu_{nf} = \mu_{bf}(1 + 1.5\phi)e^{(\frac{\phi}{1-\phi m})}$ | • Spherical <br> • $\varphi_P \ge 2$ |

Similar to thermal conductivity, many empirical equations have been developed to describe the behavior of nanofluid viscosity. Most equations are valid only for a certain range of particle sizes and concentrations. The models proposed by Corcione [96] and Sharma [68], as follows, are valid for a sufficient range of parameters:

Corcione model : $\frac{\mu_{nf}}{\mu_{bf}} = [\frac{1}{1 - 34.87(d_{np}/d_{bf})^{-0.3}\phi^{1.03}}, d_{bf} = (\frac{6M}{N\pi\rho_{f0}})^{1/3}$

M $= $ the molecular weight of the base fluid $(kg\ mol^{-1})$, N$= $ the Avogadro number
$\rho_{f0} = $ the mass density of the base fluid calculated at temperature $T_0 = 293$ K

Sharma model : $\frac{\mu_{nf}}{\mu_{bf}} = C_1(1 + \frac{\phi}{100})^{11.3}(1 + \frac{T_{nf}}{70})^{-0.038}(1 + \frac{d_{np}}{170})^{-0.061}$

where values of the coefficient $C_1$ in the Sharma model are 1.4 for SiC nanofluid and 1.0 for other nanofluids.

## 4.7. Thermophysical Property Models of Base Fluids for FPCs

In all the aforementioned equations in the preceding section, the thermophysical properties of base fluid are required to be evaluated as well. The common base liquids are water (Wat), ethylene glycol (EG), and propylene glycol (PG). The existing relevant models for water as base liquid are presented in Table 6. The data for glycol properties can be obtained from the ASHRAE standard handbook [97] via least square fitting. For instance, the equations for temperature-dependent properties at 40 wt.% PG are given in Table 6. The relations for EG-40 and EG-60 wt.% base liquid are presented in this table as well.

**Table 6.** Models for properties of base fluids.

| Model | Base Fluid | Ref. |
|---|---|---|
| **(a) Conductivity** | | |
| $k_W = \left[1.488445 + 4.12292\left(T_{avg}/298.15\right) - 1.63866\left(T_{avg}/298.15\right)\right] \times 0.6065$ | Water | [98] |
| $k_W = [0.56112 + 0.00193 \times T_W - 2.60152749e^{-6} \times T_W^2 - 6.08803e^{-8} \times T_W^3$ | Water | [68] |
| $k_{PG40} = 0.419 + 2.032 \times 10^{-4}T$ | PG40 | [99] |
| $k_{EG40} = 0.39441 + 0.00112T - 5.00323 \times 10^{-6}T^2$ | EG40 | [100] |
| $k_{EG60} = 0.33944 + 1.11 \times 10^{-3}T - 1.00528 \times 10^{-5}T^2 - 3.77393 \times 10^{-8}T^3$ | EG60 | [101] |
| **(b) Viscosity** | | |
| $\mu_W = 2.414E^{-5} \times 10^{247.8/(T_{avg}-140)}$ | Water | [85] |
| $\mu_W = [0.00169 - 4.25263e^{-5} \times T_W + 4.9255e^{-7} \times T_W^2 - 2.0993504e^{-9} \times T_W^3$ | Water | [68] |
| $\mu_{PG40} = 0.0016\left(\frac{T+273.15}{333.15}\right)^{-8}$ | PG40 | [99] |
| $\mu_{EG40} = 0.00492 - 1.24056 \times 10^{-4}T + 1.35632 \times 10^{-6}T^2 - 5.56393 \times 10^{-9}T^3$ | EG40 | [100] |
| $\mu_{EG60} = 0.00870 - 2.45439 \times 10^{-4}T + 2.82043 \times 10^{-6}T^2 - 1.178 \times 10^{-8}T^3$ | EG60 | [101] |
| **(c) Density** | | |
| $\rho_W = 1000 \times [1 - \frac{(T_W-4)^2}{119,000+1365 \times T_W - 4 \times T_W^2}]$ | Water | [68] |
| $\rho_{PG40} = 1042 - 0.479T - 0.00185T^2$ | PG40 | [99] |
| $\rho_{EG40} = 1066.79734 - 0.3071T - 0.00243T^2$ | EG40 | [100] |
| $\rho_{EG60} = 1090.6 - 0.32857T - 0.00286T^2 + 5.421 \times 10^{-19}T^3$ | EG60 | [101] |
| **(d) Heat capacity** | | |
| $C_{p,W} = [4217.629 - 3.20888 \times T_W + 0.09503 \times T_W^2 - 0.00132 \times T_W^3 + 9.415e^{-6} \times T_W^4 - 2.547e^{-8} \times T_W^5$ | Water | [68] |
| $C_{p,PG40} = 3721 + 1.629T + 0.0101T^2$ | PG40 | [99] |
| $C_{p,EG40} = 3401.21248 + 3.3443T + 9.04977 \times 10^{-5}T^2$ | EG40 | [100] |
| $C_{p,EG60} = 3044.135 + 4.2808T - 0.00186T^2 + 1.55759 \times 10^{-5}T^3$ | EG60 | [101] |

## 4.8. Heat Transfer Correlations of Nanofluid for FPCs

Heat transfer in fluids is very dependent on the flow regime, determined by Reynolds number, and the mechanism of heat transfer, i.e., forced or natural convection, which is why many investigations have focused on heat transfer performance in laminar and turbulent flows. They have followed the general form of existing correlations for heat transfer in water flow expressed based and Reynolds and Prandtl numbers. The most common models for Nusselt number that have been used for both water and nanofluids are introduced in Table 7.

**Table 7.** Correlations for the estimation of Nusselt number (Nu) in forced convection heat transfer.

| Model | Ref. |
|---|---|
| Churchill–Ozoe model for developing and developed laminar flow : $$\frac{Nu_x}{4.364[1+(Gz/29.6)^2]^{1/6}} = [1+(\frac{Gz/19.04}{[1+(Pr/0.0207)^{2/3}]^{0.5}[1+(Gz/29.6)^2]^{1/3}})^{3/2}]^{1/3}$$ | [102] |
| Churchill model for average Nusselt number in laminar flow under $T = $ cte : $$Nu_L = 3.657[1+(\frac{RePr(D/L)}{7.6})^{8/3}]^{1/8}$$ | [102] |
| Churchill model for average Nusselt number in laminar flow under $q = $ cte : $$Nu_L = 4.364[1+(\frac{RePr(D/L)}{7.3})^2]^{1/6}$$ | [102] |
| $Nu = 3.66$ fully developed laminar flow under $T = $ cte <br> $Nu = 4.36$ fully developed laminar flow under $q = $ cte | [103,104] |
| $$Nu = Nu_\infty + \frac{a(RePr\,D_h/L)^m}{1+b(RePr\,D_h/L)^n};$$ Heaton and Goldberg model for developing laminar flow <br> $a, b, m, n$ : constsnts | [105] |
| $Nu = 0.023(Re)^{0.8}(Pr)^{0.4}$ : Dituss–Boelter model $\begin{cases} and\ Re < 10,000 \\ and\ 0.6\ Pr\ 160 \end{cases}$ <br> $Nu = \frac{(f/8)(Re-1000)Pr}{1+12.7\sqrt{\frac{f}{8}}(Pr^{\frac{2}{3}}-1)}$ Petukhov model $\begin{cases} and\ 3000 < Re < 5\times10^5 \\ and\ 0.5 < Pr < 2000 \end{cases}$ <br> $f = (0.79\ln(Re) - 1.64)^{-2}$ : Petukhov model for smoothtubes <br> $1/\sqrt{f} = -2\log(\frac{\frac{\varepsilon}{d_i}}{3.7} + \frac{2.51}{Re\sqrt{f}})$ : Colebrook model for roughtubes | [106,107] |
| $Nu = \frac{(\frac{f}{8})(Re-1000)Pr}{1+12.7\sqrt{\frac{f}{8}}\cdot(Pr^{\frac{2}{3}}-1)}(\frac{Pr}{Pr_f})^{0.11}[1+(\frac{d}{L})^{\frac{2}{3}}]$ : Gnielinski model <br> $\begin{cases} 2300 < Re < 5\times10^6 \\ 0.5 < Pr < 10^6 \end{cases}$ <br> $f = (1.82\ln Re - 1.64)^{-2}$ | [108] |
| $Nu = 0.012(Re^{0.87} - 280)Pr^{0.4}$ : Gnielinski model <br> $1.5 \le Pr \le 500,\ 3000 \le Re \le 10^6$ | [44] |
| $Nu = \frac{(f/2)(Re-1000)Pr}{1+12.7\sqrt{\frac{f}{2}}(Pr^{\frac{2}{3}}-1)}$ Gnielinski model, $Re>2300$ <br> $f = \frac{0.25}{(0.79\ln Re - 1.64)}$ | [109,110] |
| $Nu = \frac{(\frac{f}{2})Re\ Pr}{1.07+\frac{900}{Re}-\frac{0.63}{1+10Pr}+12.7\sqrt{\frac{f}{2}}\cdot(Pr^{0.67}-1)}$ : Petukhov or Shah model <br> For developing flow and $Re > 10,000$ <br> $f = 0.00128 + 0.1143Re^{-0.311}$ | [111,112] |
| $Nu = \frac{(f/2)(Re-1000)Pr}{1+12.7\sqrt{f/2}(Pr^{0.67}-1)}\ 2300 < Re < ^4$ <br> $f = (158\log(Re) - 3.28)^{-2}$ | [113,114] |
| Choi model : <br> $Nu = 0.000972Re^{1.17}Pr^{1/3}$ : $Re < 2000$ <br> $Nu = 3.82\times10^{-6}Re^{1.96}Pr^{1/3}$ : $2500 < Re < 20,000$ | [115] |
| Li and Xuan model : <br> $Nu = 0.4328(1 + 11.285\phi^{0.754} Pe_{0.218})Re^{0.333}Pr^{0.4}$ : laminar flow$(Re \le 2300)$ <br> $Nu = 0.0059(1 + 7.6286\phi^{0.6886} Pe_{0.001})Re^{0.9238}Pr^{0.4}$ : turbulent flow$(Re > 2300)$ | [58,116] |

**Table 7.** *Cont.*

| Model | Ref. |
|---|---|
| Sharma model for turbulent forced convection : $$\mathrm{Nu} = 0.023 Re^{0.8} Pr_w^{0.4} (1 + Pr_{nf})^{-0.012} (1 + \varphi)^{0.23}$$ | [68] |
| Buongiono model for CuO turbulent flow : $$\mathrm{Nu}_b = \frac{f/8(Re_b - 1000)Pr_b}{1 + \delta_v^+ \sqrt{f/8}(Pr_v^{2/3} - 1)}$$ $b = \text{bulk}, \delta_v^+ = \text{laminar sublayer dimensionless thickness}$ | [117] |
| Vajjha model for turbulent, $W : EG(40 : 60), CuO, SiO_2$ and $Al_2O_3$ $$\mathrm{Nu} = 0.065(Re^{0.65} - 60.22)(1 + 0.0169\phi^{0.15})Pr^{0.542}$$ $$f = 0.3164 Re_{-0.25}(\tfrac{\rho_{nf}}{\rho_{bf}})^{0.797}(\tfrac{\mu_{nf}}{\mu_{bf}})^{0.108}$$ | [44] |
| Maiga model for turbulent flow : $$\mathrm{Nu} = 0.085 Re^{0.71} Pr^{0.35}$$ | [58,118] |
| Maiga model for laminar flow, $Re \le 1000$ : $\mathrm{Nu} = 0.086 Re^{0.55} Pr^{0.5}$: constant wall flux $\mathrm{Nu} = 0.28 Re^{0.35} Pr^{0.36}$: constant wall temperature | [58,119] |
| Pack and Cho model for turbulent flow : $$\mathrm{Nu} = 0.021 Re^{0.8} Pr^{0.5}$$ | [49] |
| Bejan model for turbulent flow : $$\mathrm{Nu} = 0.021(Re^{0.87} - 280)Pr^{0.4}$$ | [120] |
| Anoop model for laminar flow : $$\mathrm{Nu}_x = 4.36 + \left[6.219 \times 10^{-3} x_+^{1.1522}(1 + \phi^{0.1533})e^{-2.5228x_+}\right]\left[1 + 0.57825\left(\tfrac{d_p}{d_{ref}}\right)^{-0.2183}\right]$$ $d_{ref} = 100nm, x_+ = \text{dimensionless distance}$ | [121] |
| Chandrasekar model for developed laminar flow of $Al_2O_3$ under $q = $ cte with and without wire coil insert : $$\mathrm{Nu}_x = 0.279(RePr)^{0.558}(\tfrac{p}{d})^{-0.447}(1 + \phi)^{134.65}$$ $$f = 530.8 Re^{-0.909}(\tfrac{p}{d})^{-1.388}(1 + \phi)^{-512.26}$$ | [121] |
| Suresh model for developed turbulent flow of CuO under $q = $ cte in a dimpled tube $$\mathrm{Nu}_x = 0.00105 Re^{0.984} Pr^{0.4}(1 + \phi)^{-80.78}(1 + \tfrac{p}{d})^{2.089}$$ $$f = 0.1648 Re^{0.97}(1 + \phi)^{107.89}(1 + \tfrac{p}{d})^{-4.463}$$ | [121] |
| Suresh model for laminar flow in a tube with helical screw tape insert : $\mathrm{Nu} = 0.5419(RePr)^{0.53}(\tfrac{p}{d})^{0.594}$ : $Al_2O_3$ $\mathrm{Nu} = 0.5657(RePr)^{0.5337}(\tfrac{p}{d})^{0.6062}$ : CuO | [121] |
| Hojjat model for turbulent non $-$ Newtonian fluid flow in a heated tube : $$\mathrm{Nu} = 7.135 \times 10^4 Re^{0.9545} Pr^{0.913}(1 + \phi^{0.1358})$$ | [121] |
| Hojjat model for turbulent non-Newtonian fluid flow in a concentric tube : $$\mathrm{Nu} = 0.0115 Re^{1.050} Pr^{0.693}(1 + \phi^{0.388})$$ | [121] |
| Eiamsa-ard and Kiatkittipong model in twisted insert tube : $$\mathrm{Nu} = c_1(1 + N)^{c_2}(1 + \phi)^{c_3} Re^{c_4} Pr^{c_5}$$ $$f = c_6(1 + N)^{c_7}(1 + \phi^{0.388})^{c_8} Re^{c_9}$$ $c_1 \text{ to } c_9 \text{ constants for different categories of inserts}, N = \text{number of inserts}$ | [122] |

Attention must be given to the conditions for which the above correlations are valid. The range of different parameters, flow regime, the type of fluid and material are important factors to be considered before selection of an appropriate correlation for when evaluating heat transfer coefficient in a nanofluid-based solar FPC.

## 5. Thermal Efficiency of Nanofluid-Based FPCs

When treating the thermal efficiency of a nanofluid-based FPC, which is governed by factor $F'$ [123], the internal heat transfer coefficient $h_{fi}$ plays its role and is affected by the nanofluid used.

$$F' = \frac{1/U_L}{W[1/[U_L(D + (W-D)F] + 1/C_b + 1/\pi D_i h_{fi}]} \tag{9}$$

where $U_L$ is overall heat loss coefficient, $W$, $D$, $F$ are geometrical parameters of the FPC, and $C_b$ is bond conductance (see [123]). Heat transfer coefficient is related with Nusselt number, Nu, based on the following equation, taking the conductivity of nanofluid $k_{nf}$ into account:

$$h_{fi} = \frac{\text{Nu}k_{nf}}{D_i} \tag{10}$$

Once the flow regime is determined using Reynolds number (Re) in Equation (11), the appropriate correlation for the Nusselt number given in Table 7 might be chosen for heat transfer assessment in a nanofluid-filled FPC.

$$\text{Re}_{\text{nf}} = \frac{\rho_{nf}VD_i}{u_{nf}} = \frac{4\dot{m}_{riser}}{\pi D_i u_{nf}} \tag{11}$$

where, $\dot{m}_{riser}$ is the mass flow rate of the collector riser tube. The relevant model for the estimation of nanofluid viscosity, density, and other thermophysical properties can be selected from Table 6.

Figures 8 and 9 show an interesting result for the trends of Nu number and heat transfer coefficient. Although the Nusselt number in a nanofluid-based FPC decreases with nanofluid concentration, the heat transfer coefficient enhances conversely. This refers to the fact that the rate of enhancement in conductivity overcomes the rate of decrease in Nusselt number. Therefore, it is expected the efficiency of a nanofluid-based FPC to be improved with nanofluid volume concentration.

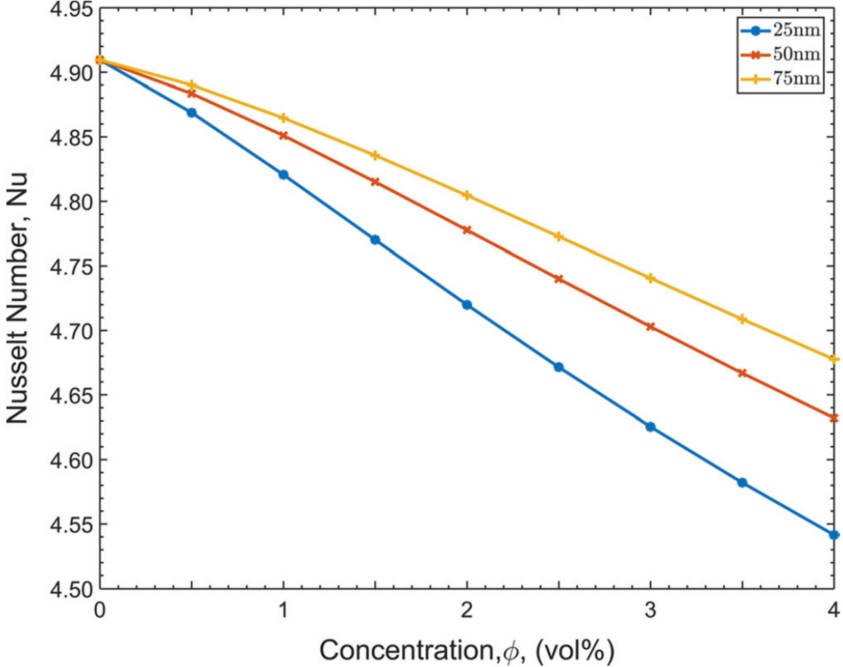

**Figure 8.** Variation of Nu number in a Cu-nanofluid-based FPC [124].

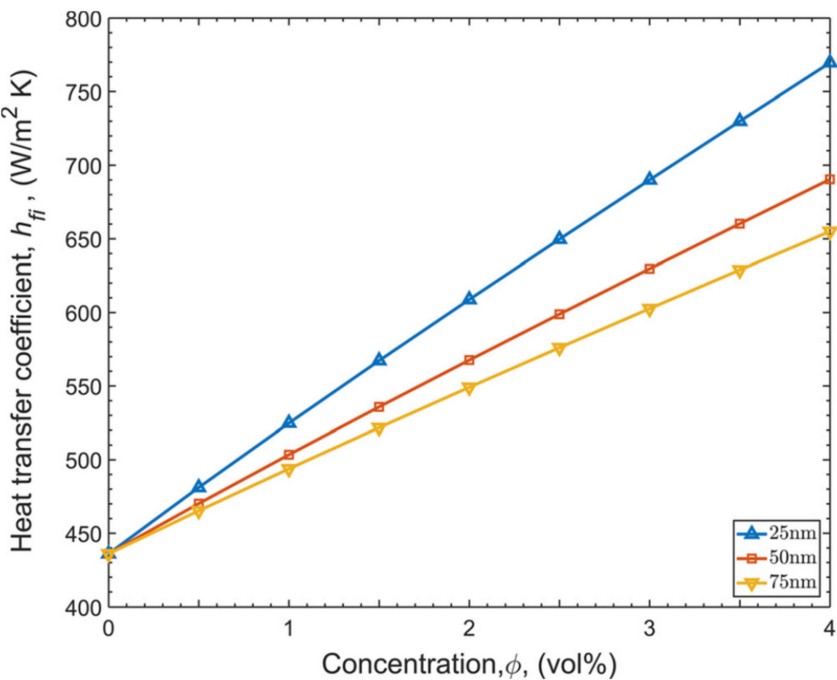

**Figure 9.** Variation of heat transfer coefficient in a Cu-nanofluid-based FPC [124].

The effect of particle size is also illustrated in both figures. As seen, smaller nanoparticles cause the Nusselt number to decrease and heat transfer coefficient to increase due to the conductivity enhancement by smaller particles. It is shown that nickel amongst metallic nanoparticles and cerium oxide between oxide nanoparticles perform better than the other commonly-used nanomaterials in FPCs [33]. Altering nanofluid pH with respect to the isoelectric point tends to increase the positive influence of nanofluid on the efficiency of a nanofluid-based FPC [125]. Nanoparticles with platelet shapes exhibit the lowest effect on the efficiency enhancement of a collector, while the brick-shaped particles cause the highest effect on efficiency improvement [126].

Thermal efficiency of FPC is associated with how it absorbs the radiation heat and how it loses energy. The main equation for the thermal efficiency of an FPC is written as follows:

$$\eta_{th} = \eta_I = \frac{Q_u}{A_c G_t} = \frac{\dot{m} C_p \left( T_{f,out} - T_{f,in} \right)}{A_c G_t} \tag{12}$$

The history of using a mixture of water and another additive in a solar collector dates back to 1975 [74]. In fact, water soluble dyes called "black liquid" were exploited as a new working fluid in a direct absorption solar (DAC) FPC, and the results showed an enhancement in collector heat loss. After developing the concept of using particle suspensions in HTF of solar collectors in 1979 [127], 1990 [128], 2000 [129], and 2004 [130], a nanofluid-laden DAC was tested by Tyagi et al. [131] in 2009. They used water–aluminum (Al) nanofluid to evaluate analytically the performance of the DAC. Their results demonstrated that radiation adsorption increased nine times than that with pure water. This caused the efficiency of a DAC FPC to increase by 10% compared with a conventional water-based FPC. Hereafter, investigations on the application of nanofluids in FPCs became more attractive to open a new field of study. A detailed summary of the research works carried out to utilize different nanofluids in FPCs is given in Table 8, which can be used as a quick reference to address the major topics on which the investigations focused. A part of this table represents the selected base liquid in different studies. Although the majority of researchers selected water-based nanofluids, there are studies based on mixtures of water and glycols, i.e., W:EG, W:PG, and W:PEG. Glycols can be used in FPCs either as antifreeze or as stabilizer of nanofluid. However, they are chemically-derived products or even toxic in

the case of EG. Therefore, they cannot be considered as environmentally-friendly chemicals and cannot assist environmental sustainability. Recently, bio-glycol (BG) which is a renewable (corn)-based glycol product has been suggested to be applied in FPCs [132]. The performance of a Cu nanofluid-charged FPC with different base liquids, namely W:BG, W:PG, and W:EG, were compared, as shown in Figure 10. As it is observed, the efficiency of the collector working with BG-based copper nanofluid was even a bit higher than that with the PG and EG case.

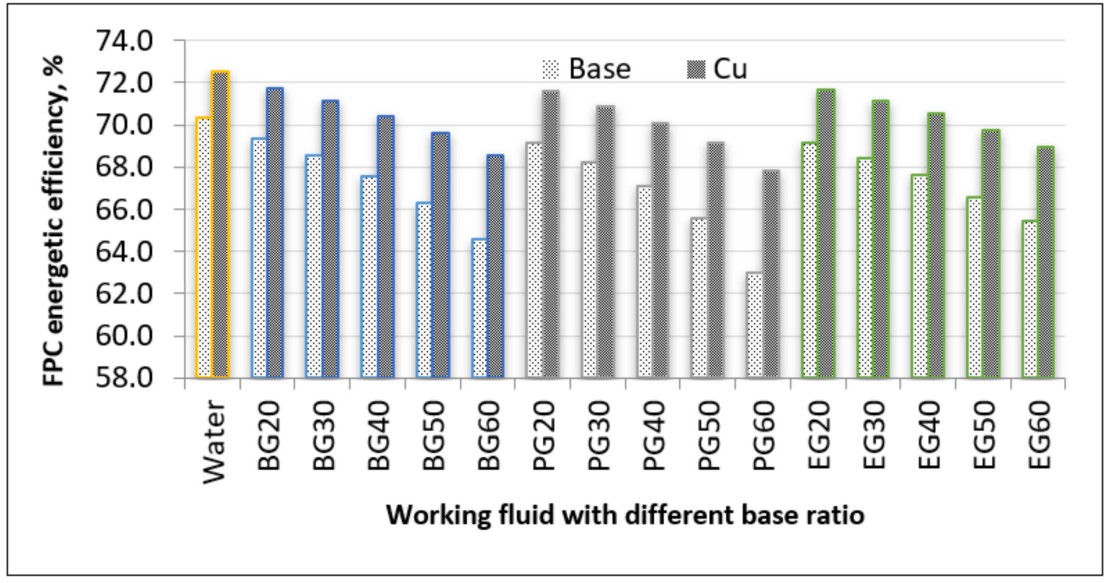

**Figure 10.** Energy efficiency of a Cu nanofluid-based FPC vs. base ratio of glycol products.

Moreover, it was found that the lower base ratios of water and BG mixture provided higher efficiencies. Therefore, bio-glycol (BG) could be considered as a good alternative for antifreeze or thermal fluid in solar FPCs.

## 6. Exergy Efficiency of Nanofluid-Based FPCs

FPC is a major part of a solar water heater (SWH), and thus its optimum performance is very crucial. However, the first-law or energy or thermal efficiency of a collector cannot be a good enough tool for the thorough assessment of system losses [133], which is why second-law or exergy efficiency is applied to the system to recognize all the bottlenecks in terms of inefficiencies, their magnitude, and the places they occur. Therefore, both first-law and second-law analyses are applied for the performance assessment of FPCs and for determining the optimum range of operating conditions. In fact, the simultaneous application of both laws' analyses provides capable criteria for qualitative and quantitative performance analysis of FPCs [134]. The exergy analysis for conventional solar collectors had been reported over the years when Bejan et al. [135] in 1981 carried out a second-law analysis using the Carnot model of incident exergy. The exergy of the sun, according to the Carnot model, is estimated based on ambient temperature $T_a$, apparent temperature of the sun $T_s$ as an exergy source, and solar irradiation $Q$ falling on the collector:

$$Ex_{sun} = Q(1 - \frac{T_a}{T_s}) \tag{13}$$

The value of $T_s$ is considered to be 4500 K. The optimum operating conditions for minimum heat transfer irreversibility or maximum exergy delivery were derived. It should be noted that the Carnot model was introduced of the first time by Jeter [136]. Later, Suzuki [137], in 1988, presented a fundamental model for exergy balance of solar collectors using the Carnot model as exergy of solar radiation. Kar [138], in 1989, determined optimum mass flow through a conventional FPC using

the exergy maximization method. He considered the Carnot equation for the exergy of the sun and maximum heat collection and minimum pumping power for the optimization. Exergy analysis for conventional FPCs has been carried out by many researchers [139–142]. Ajam et al. [143], in 2005, claimed that another famous model developed by PLP for the exergy of solar radiation violates the second-law requirements. The PLP model is written as follows:

$$Ex_{sun} = Q[1 - \frac{4}{3}\frac{T_a}{T_s} + \frac{1}{3}(\frac{T_a}{T_s})^4] \tag{14}$$

They preferred to utilize the Carnot model in their analysis. Other researchers [144–152] performed the second-law analysis for conventional FPCs using either the Carnot or PLP model for the radiation exergy coming from the sun. However, in the case of nanofluid-based FPCs, second-law analysis was accomplished analytically for the first time by [134]. It should be noted that the second term in Equations (13) and (14) denotes the maximum exergy of solar incident and it is called the exergy factor, $\eta_{max,s}$. Therefore, it can be estimated based on Carnot or PLP model.

A comprehensive comparative summary of research investigations carried out for thermal performance of nanofluid-filled FPCs is given in Table 8 in a chronological manner. The table compares the different performance criteria on which the studies focused, and corresponding main achievements are stated. The criteria might be first-law efficiency, $\eta_I$, second-law efficiency, $\eta_{II}$, Nusselt number, Nu, convective heat transfer coefficient, $h_{fi}$, outlet temperature, $T_{out}$, rate of generated entropy, $\dot{S}_{gen}$, rate of heat transfer, Q, nanofluid properties, etc. The value of absorber plate emittance, $\varepsilon_p$, and absorptance, $\alpha_p$, sun temperature, $T_{sun}$, and temperature difference between collector inlet and ambient air, $\Delta T_{in\text{-}air}$, and employed exergy factor, $\eta_{max,s}$ model are represented. The table shows whether the exergy destruction due to pressure loss, $\dot{E}_{des,\Delta p}$, and pressure drop in collector headers, $\Delta p_{header}$, have not been considered (marked as "NC"); are not specified (marked as "NS") by the researchers; or even are not applicable (marked as "NA"). The nanofluid flow regime model utilized for the estimation of friction factor, $f$; Nusselt number, Nu; conductivity, $k$; and viscosity, $\mu$, of nanofluids are compared in different studies. The type of selected base fluid (BF), nanoparticle (NP), and surfactant along with concentration (vol.%, wt.%, or ppm) and size, $d_p$, are presented for each study. The base liquid includes water (Wat), ethylene glycol (EG), propylene glycol (PG), polyethylene glycol (PEG), or a mixture of water and glycol products (W:EG, W:PG, etc.).

**Table 8.** Comparative summary of research investigations for thermal performance of nanofluid-filled FPCs. *.

| Ref. | Performance Criterion | $\varepsilon_p$ (%) | $\alpha_p$ (%) | $\eta_{max,s}$ | $T_{sun}$ (K) | $E_{des,\Delta p}$ | $\Delta p_{header}$ | $\Delta T_{in-air}$ | Flow Regime | Nu Model | $f$ Model | NP Type | BF Type | $k$ Model | $\mu$ Model | Concentration | $d_p$ (nm) | Surfactant | Research Technique | Main Achievements |
|---|---|---|---|---|---|---|---|---|---|---|---|---|---|---|---|---|---|---|---|---|
| [131] | $\eta_I$ | NS | NS | NA | 5800 | NA | NA | 0 | NS | NA | NA | Al | Wat | NS | NS | 0.8 vol.% | 5 | NA | Num | +10% in $\eta_I$ |
| [153] | $k$ | NS | NS | NA | NS | NA | NA | NA | NA | NA | NA | MWCNT | Wat | Hamilton–Crosser | NS | 1.0 vol.% | 283 | SDS | Exp | +41% in $k$ |
| [154] | $\eta_I$ | NS | NS | NA | NS | NA | NA | NS | NS | NA | NA | CNT, Graphite, silver | Wat | NS | NA | 0 to 1.0 vol.% | 6–20, 30, 20, 40 | | Ana and Exp | +6% in $\eta_I$ |
| [155] | $\eta_I$ | 7 | 96.2 | NA | NS | NA | NA | NA | NS | NA | NA | Al$_2$O$_3$ | Wat | NS | NA | 0.2 wt.% and 0.4 wt.% | 15 | Triton X-100 | Exp | +28.3% in $\eta_I$ |
| [156] | $\eta_I$ | 7 | 96.2 | NA | NS | NA | NA | NA | NS | NA | NA | MWCNT | Wat | NS | NA | 0.2 wt.% and 0.4 wt.% | 10–30 | Triton X-100 | Exp | decrease in $\eta_I$ at 2 wt.% when not using surfactant |

**Table 8.** *Cont.*

| Ref. | Performance Criterion | $\varepsilon_p$ (%) | $\alpha_p$ (%) | $\eta_{max,s}$ | $T_{sun}$ (K) | $E_{des,\Delta p}$ | $\Delta p_{header}$ | $\Delta T_{in\text{-}air}$ | Flow Regime | Nu Model | $f$ Model | NP Type | BF Type | $k$ Model | $\mu$ Model | Concentration | $d_p$ (nm) | Surfactant | Research Technique | Main Achievements |
|---|---|---|---|---|---|---|---|---|---|---|---|---|---|---|---|---|---|---|---|---|
| [157] | Nu | NS | NS | NA | NS | NA | NC | NS | laminar | Lin and Violi | NA | Ag, CuO | Wat | NS | NA | 0–15 vol.% | NS | NA | Num | $Nu_{Al2O3} > Nu_{CuO}$ |
| [134] | $\eta_I, \eta_{II}$ and $h_{fi}$ | NS | NS | Carnot | 4350 | NC | NC | NS | laminar | Choi | Darcy | Al$_2$O$_3$, TiO$_2$, SiO$_2$, CuO | Wat | Hamilton–Crosser | NS | 0–4 vol.% | NS | NA | Ana | $\eta_{I,CuO} >$ others +22.15% in $h_{fi}$ −4.34% in $S_{gen}$ |
| [158] | $\eta_I$ | NS | NS | NA | NS | NA | NA | NA | NS | NA | NS | Cu | Wat | NS | Einstein | 0.05 and 0.1 wt.% | 35 | SDS | Exp | +24% in $\eta_I$ |
| [159] | $\eta_I$ | <10 | >95 | NA | NS | NA | NA | NA | NS | NA | NA | CNT | Wat | NS | NS | 0.4–0.6, 1.0 wt.% | 1 | Polysorbate80 | Exp | +39% in $\eta_I$ |
| [160] | $\mu, \rho,$ and $\Delta p$ | NS | NS | NA | 4350 | NA | NC | NS | Laminar | NA | Darcy | Al$_2$O$_3$ | W:EG | Measured | Measured | 0.05 to 0.1 vol.% | 13 | NS | Theo | Insignificant change in pumping power |
| [161] | $k$ $V_{sed}$ | NS | NS | NA | NS | NA | NC | NS | Laminar and turbulent | Measured | NS | Al$_2$O$_3$, Fe$_2$O$_3$ and Al$_2$O$_3$ | Wat | Measured | Brinkmann | 1–3 vol.% | 45, 30 and 60 | NS | Num and Exp | +6.7% in $k$ header modification |

**Table 8.** *Cont.*

| Ref. | Performance Criterion | $\varepsilon_p$ (%) | $\alpha_p$ (%) | $\eta_{max,s}$ | $T_{sun}$ (K) | $E_{des,\Delta p}$ | $\Delta p_{header}$ | $\Delta T_{in\text{-}air}$ | Flow Regime | Nu Model | $f$ Model | NP Type | BF Type | $k$ Model | $\mu$ Model | Concentration | $d_p$ (nm) | Surfactant | Research Technique | Main Achievements |
|---|---|---|---|---|---|---|---|---|---|---|---|---|---|---|---|---|---|---|---|---|
| [162] | Nu | NS | NS | NA | NS | NA | NS | NA | Laminar | Shah | Darcy | $Al_2O_3$ | Wat | Sharma | Sharma | 0.02, 0.1, and 0.5 vol.% | NS | NS | Exp | +8–12% in Nu |
| [163] | $\eta_I$ | NS | NS | NA | NS | NA | NA | NA | Laminar and turbulent | NA | NS | $TiO_2$ | Wat | NS | NS | 0–0.3 wt.% | 20 | NU | Exp | Lower flow rate caused higher enhancement. Surfactant decreased efficiency. |
| [164] | $\eta_I$ and $h_{fi}$ | 5 | 96 | NA | NS | NA | NA | NS | Laminar | NS | NA | MWCNT $Al_2O_3$ CuO | Wat | Maxwell | Einstein | 1, 2, and 3 wt.% | NS | NA | Num (CFD) | $\eta_{I,CuO}$ > others |
| [165] | $\eta_I$ | 7 | 96.2 | NA | NS | NA | NA | 0 to 90% | NS | NS | NA | CuO | Wat | NS | NS | 0.4 vol.% | 40 | NS | Exp | +21.8% in $\eta_I$ |
| [166] | $\eta_I$ | NS | NS | NA | NS | NA | NA | NA | NS | NS | NA | Cu + Ag | W:EG | NS | NS | 0.3 vol.% | 7 | NS | Exp | +3% in $\eta_I$ |
| [167] | $\eta_I$ | NS | NS | NA | NS | NA | NA | NA | NS | NS | NA | Cu | W:EG | NS | NS | 0.3 vol.% | 7 | NS | Exp | +3.2% in $\eta_I$ |

**Table 8.** *Cont.*

| Ref. | Performance Criterion | $\varepsilon_p$ (%) | $\alpha_p$ (%) | $\eta_{max,s}$ | $T_{sun}$ (K) | $E_{des,\Delta p}$ | $\Delta p_{header}$ | $\Delta T_{in\text{-}air}$ | Flow Regime | Nu Model | $f$ Model | NP Type | BF Type | $k$ Model | $\mu$ Model | Concentration | $d_p$ (nm) | Surfactant | Research Technique | Main Achievements |
|---|---|---|---|---|---|---|---|---|---|---|---|---|---|---|---|---|---|---|---|---|
| [168] | $\eta_I$ | NS | NS | NA | NS | NA | NA | NA | NS | NS | NA | Cu | W:EG | Measured | Einstein | 0.2 and 0.3 wt.% | 10 | NS | Exp | +10% in $\eta_I$ |
| [169] | $\eta_I$ | NS | NS | NA | NS | NA | NA | NS | NS | NS | NA | Ag, Cu, Al$_2$O$_3$, CuO | Wat | Maxwell | Pak and Cho | 0–10% | NS | NA | Num | $\eta_{I, Ag} >$ others $0278_{opt} = 5\%$ |
| [170] | Nu and $k, \dot{S}_{gen}$ | 92 | NS | Carnot | 4350 | C | NC | 0 | Turbulent | Gnielinski | Colebrook | Al$_2$O$_3$, TiO$_2$, SiO$_2$, Cu | Wat | Xuan | Corcione | 0–4 vol.% | 25 | NA | Ana | Nu$_{Al2O3} >$ others $T_{out,Cu} >$ others $\dot{S}_{gen,TiO2} <$ others |
| [171] | $\eta_I$ | 7 | 96.2 | NA | NS | NA | NA | NA | NS | NS | NA | NA | W:PG | NS | NA | NA | NA | NA | Exp | −15.68% in $\eta_I$ (BR = 25%) −8.3% in $\eta_I$ (BR > 75%) |
| [172] | $\eta_I$ | NS | NS | NA | NS | NA | NA | NA | Laminar and turbulent | NS | NA | CuO | Wat | Measured | Measured | 0.05 vol.% | 75 | SDBS | Exp | $\Delta\eta_{I, natural} > \Delta\eta_{I, forced}$ |
| [39] | $\eta_I, D_i, G_t$ | 5 | 95 | NA | NS | NA | NS | NS | Laminar | NS | NA | Cu | Wat | Maxwell | Pak and Cho | 2 vol.% | 5 | NA | Num (CFD) | +6% in $\eta_I$ |

**Table 8.** *Cont.*

| Ref. | Performance Criterion | $\varepsilon_p$ (%) | $\alpha_p$ (%) | $\eta_{max,s}$ | $T_{sun}$ (K) | $E_{des,\Delta p}$ | $\Delta p_{header}$ | $\Delta T_{in-air}$ | Flow Regime | Nu Model | $f$ Model | NP Type | BF Type | $k$ Model | $\mu$ Model | Concentration | $d_p$ (nm) | Surfactant | Research Technique | Main Achievements |
|---|---|---|---|---|---|---|---|---|---|---|---|---|---|---|---|---|---|---|---|---|
| [173] | $\eta_I$, pH | NS | 77 | NA | NS | NA | NA | NA | Laminar | Churchill | NA | $Al_2O_3$ CuO | Wat | Maxwell | Brinkmann | 0.1 wt.% 0.2 wt.% | 40 20 | SDS | Exp | $\eta_{I,\,Al2O3} > \eta_{I,\,CuO}$ at high pH |
| [174] | $\eta_I$, $h_{fi}$ | NS | NS | NA | NS | NA | NA | NA | Turbulent | Measured | NA | Ag | Wat | Xuan | Einstein | 0.01, 0.03, 0.04 vol.% | <100 | PVP | Exp | +18% in $h_{fi}$ |
| [175] | $\eta_I$ | NS | NS | NA | NS | NA | NC | NA | Turbulent | NS | Blasius | Cu | Wat | Measured | Measured | 0.01, 0.02, 0.04, 0.1, 0.2 wt.% | 25 and 50 | NS | Exp | +23.83% in $\eta_I$ at 0.1 wt.% |
| [176] | $Q$ | NS | NS | NA | NS | NA | NC | NA | Laminar | McAdams | NA | ZnO | PG | Measured | Measured | 0–2 vol.% | 32.4 | NU | Exp | +23.83% in $Q$ (heat transfer) |
| [177] | $\eta_I$ | NS | NS | NA | 4350 | NA | NA | NA | NS | NS | NS | $SiO_2$ | W:EG | NS | NS | 0.5, 0.75, 1.0 vol.% | 40 | NS | Exp | $\Delta\eta_I$ = +4 to +8% |

**Table 8.** *Cont.*

| Ref. | Performance Criterion | $\varepsilon_p$ (%) | $\alpha_p$ (%) | $\eta_{max,s}$ | $T_{sun}$ (K) | $E_{des,\Delta p}$ | $\Delta p_{header}$ | $\Delta T_{in-air}$ | Flow Regime | Nu Model | $f$ Model | NP Type | BF Type | $k$ Model | $\mu$ Model | Concentration | $d_p$ (nm) | Surfactant | Research Technique | Main Achievements |
|---|---|---|---|---|---|---|---|---|---|---|---|---|---|---|---|---|---|---|---|---|
| [47] | $\Delta T_{in-out}$ | NS | NS | NA | NS | NA | NA | NA | NS | NA | NA | $Al_2O_3$ | Wat | NS | NS | 0.5 vol.% | 20–40 | NS | Exp | +4 °C in $\Delta T$ |
| [178] | $\eta_I$ $k$ | 12 | 94 | Carnot | NS | NC | NC | NA | Laminar | NS | NS | $TiO_2$ | W:PLE | Measured | Measured | 0.1 and 0.3 vol.% | 21 | PEG400 | Exp | +34.5% in $\eta_I$ +6% in $k$ |
| [179] | $\eta_I$, $\eta_I$, $d_p$ and pH | 12 | 94 | Carnot | NS | NC | NC | NA | NS | NS | NS | $Al_2O_3$ | Wat | Measured | NS | 0.1 vol.% | 13 and 20 | NU | Exp | $\eta_{I, 13nm} > \eta_{I, 20nm}$ by 3% $\eta_{II, 13nm} > \eta_{II, 20nm}$ by 5% |
| [180] | $\eta_I$ | 12 | 94 | Carnot | NS | C | NC | NA | NS | NS | NS | $Al_2O_3$ | Wat | Measured | NS | 0.1 and 0.3 vol.% | 13 | NS | Exp | +83.5% in $\eta_I$ |
| [181] | $\bar{h}_{f,i}$ | NS | NS | NA | NS | NA | NA | NA | Laminar | Ranz–Marshall | NS | $TiO_2$ | Wat and W:EG | Measured | Measured | 2.3 vol.% | 21 | CTAB | Num and Exp | +21% in $\bar{h}_{f,i}$ |
| [182] | $\eta_I$ | NS | NS | NA | NS | NA | NA | NA | NS | NA | NA | $Al_2O_3$ | Wat | NS | NS | 3.0 vol.% | 45 | NS | Exp | +7% in $\eta_I$ |

**Table 8.** *Cont.*

| Ref. | Performance Criterion | $\varepsilon_p$ (%) | $\alpha_p$ (%) | $\eta_{max,s}$ | $T_{sun}$ (K) | $E_{des,\Delta p}$ | $\Delta p_{header}$ | $\Delta T_{in\text{-}air}$ | Flow Regime | Nu Model | $f$ Model | NP Type | BF Type | $k$ Model | $\mu$ Model | Concentration | $d_p$ (nm) | Surfactant | Research Technique | Main Achievements |
|---|---|---|---|---|---|---|---|---|---|---|---|---|---|---|---|---|---|---|---|---|
| [183] | $\eta_I$ | 92 | 95 | NA | NS | NA | NA | NA | NS | NA | NA | MWCNT | W:EG | NS | NS | 0–100 ppm | NS | NS | Exp | +23% in $\eta_I$ |
| [184] | $\eta_I$ and $T_{out}$ | NS | 97 | NA | NS | NA | NA | NA | Laminar | Goldberg | NA | Graphene | Wat | Measured Xuan | Measured Brinkmann | 0.01 and 0.02 wt.% | 265 | NS | Exp and Ana | +18.87% in $\eta_I$ +14 °C in $T_{out}$ |
| [185] | $\eta_I$ | NS | NS | NA | NS | NA | NC | NA | Laminar | Shah, Churchill, and Sieder | Darcey | Graphene oxide | Wat | Measured | Measured | 0.005, 0.01, 0.02 wt.% | 300 | NU | Exp | +7.3% in $\eta_I$ |
| [186] | $\eta_I$ | 7 | 96.2 | NA | NS | NA | NA | NA | NS | NA | NA | $Al_2O_3$ | Wat | NS | NS | 0.15 wt.% | 20 | Triton X-100 | Exp | +18% in $\eta_I$ |
| [187] | $\eta_I$ and $T_{out}$ | NS | NS | NA | NS | NA | NA | NA | Turbulent | NA | NA | Ag | Wat | NS | NS | 0.01, 0.03, and 0.04 vol.% | NS | NA | Num and Exp | Error = ±2% |

**Table 8.** *Cont.*

| Ref. | Performance Criterion | $\varepsilon_p$ (%) | $\alpha_p$ (%) | $\eta_{max,s}$ | $T_{sun}$ (K) | $E_{des,\Delta p}$ | $\Delta p_{header}$ | $\Delta T_{in\text{-}air}$ | Flow Regime | Nu Model | $f$ Model | NP Type | BF Type | $k$ Model | $\mu$ Model | Concentration | $d_p$ (nm) | Surfactant | Research Technique | Main Achievements |
|---|---|---|---|---|---|---|---|---|---|---|---|---|---|---|---|---|---|---|---|---|
| [188] | $\eta_I$ | NS | NS | NA | NS | NA | NA | NA | NS | NA | NA | Al$_2$O$_3$ | Wat | Measured | Chen | 0–3 vol.% | 45 | NA | Num | +7.54% in $\eta_I$ |
| [189] | $\eta_I$ and $\eta_{II}$ | 12 | NS | Carnot and PLP | NS | C | NC | NA | Laminar | NA | Darcey | MWCNT, Graphene, CuO, Al$_2$O$_3$, TiO$_2$, and SiO$_2$ | Wat | Measured | Measured | 0–2.25 vol.% | 7, 20, 42, 45, 44,and 10 | Triton X-100 | Exp | +23.47% in $\eta_I$ +29.32% in $\eta_{II}$ |
| [190] | $\phi$ and $h_{fi}$ | NS | NS | NA | NS | NA | NC | NA | Laminar | Cerón | Darcey | Al$_2$O$_3$ | Wat | Xuan | Maiga | 0–4 vol.% | 25 and 100 | NA | Num (CFD) | +58% in $h_{fi}$ $\phi_{opt}$ = 2% |
| [191] | Nu and $T_{out}$ | NS | NS | NA | NS | NA | NC | NA | Laminar | Cerón | Darcey | Al$_2$O$_3$ | Wat | Xuan | Maiga | 0–5 vol.% | 25 | NA | Num (CFD) | Nu and $T_{out}$ decreased with $\phi$ |
| [192,193] | $\eta_I$, absorber porosity | NS | NS | Carnot | NS | C | NC | NA | Laminar | Measured | NS | SiO$_2$ | Wat | Xuan | Brinkmann | 0.2, 0.4, 0.6 vol.% | 25–30 | NS | Exp | +8.1% in $\eta_I$ |
| [194] | $Q_u$ | 7 | 96.2 | NA | NS | NA | NA | NA | NS | NA | NA | Al$_2$O$_3$CuO | Wat | Xuan | NA | 0.1 vol.% | 20 40 | NS | Exp | +29.5% in $Q_u$ |

**Table 8.** *Cont.*

| Ref. | Performance Criterion | $\varepsilon_p$ (%) | $\alpha_p$ (%) | $\eta_{max,s}$ | $T_{sun}$ (K) | $E_{des,\Delta p}$ | $\Delta p_{header}$ | $\Delta T_{in-air}$ | Flow Regime | Nu Model | $f$ Model | NP Type | BF Type | $k$ Model | $\mu$ Model | Concentration | $d_p$ (nm) | Surfactant | Research Technique | Main Achievements |
|---|---|---|---|---|---|---|---|---|---|---|---|---|---|---|---|---|---|---|---|---|
| [195] | $\eta_I$ | 7 | 96.2 | NA | NS | NA | NA | NA | NS | NA | NA | $Al_2O_3$ | Wat | Calvin–Petersona | NA | 0.1 vol.% | 20 | NS | Exp | +23.5% in $\eta_I$ |
| [196] | $\eta_I$ | 13 | 95 | NA | NS | NA | NA | NA | NS | NA | NA | $WO_3$ | Wat | Xuan | NA | 0.017, 0.033, 0.067 vol.% | 90 | NS | Exp | +13.48% in $\eta_I$ at 0.067 vol.% |
| [197] | $\eta_I$ | 13 | 95 | NA | NS | NA | NA | NA | NS | NA | NA | $CeO_2$ | Wat | Xuan | NA | 0.017, 0.033, 0.067 vol.% | 25 | NS | Exp | +10.47% in $\eta_I$ at 0.067 vol.% |
| [198] | $\eta_I$ | 95 | NS | NA | NS | NA | NA | NA | NS | NA | NA | $CeO_2$ | Wat | Xuan | Corcione | 0.01 vol.% | 25 | NS | Exp and Theo | +21.5% in $\eta_I$ |
| [199] | $\eta_I$ | NS | NS | NA | NS | NA | NA | NA | NS | NA | NA | $Al_2O_3 + TiO_2$ | Wat | NS | Brinkmann | 0.1 wt.% | 20 + 15 | CTAB | Exp and Num | +26% in $\eta_I$ |
| [200] | $\eta_I$ | NS | NS | NA | NS | NA | NA | NA | NS | NA | NA | $TiO_2$ | Wat | NS | Measured | 2 wt.% | | Triton X-100 | Exp | +12.47% in $\eta_I$ |

**Table 8.** *Cont.*

| Ref. | Performance Criterion | $\varepsilon_p$ (%) | $\alpha_p$ (%) | $\eta_{max,s}$ | $T_{sun}$ (K) | $E_{des,\Delta p}$ | $\Delta p_{header}$ | $\Delta T_{in-air}$ | Flow Regime | Nu Model | $f$ Model | NP Type | BF Type | $k$ Model | $\mu$ Model | Concentration | $d_p$ (nm) | Surfactant | Research Technique | Main Achievements |
|---|---|---|---|---|---|---|---|---|---|---|---|---|---|---|---|---|---|---|---|---|
| [201] | $Q$ (from PV) | NS | NS | NA | NS | NA | NA | NA | NS | Shah | NA | $Al_2O_3$ | Wat | Corcione | Corcione | 0–6 vol.% | 20 and 40 | NA | Num | −5 K in PV system top side temperature |
| [202] | $T_{out}$ and $\eta_{II}$ | 5 | 95 | Carnot | 4,500 | NC | NC | NS | Laminar and Turbulent | NS | Darcey | $Al_2O_3$ | Wat | Measured | Measured | 1, 2, and 3 vol.% | NS | NA | Num | +7.20% in $T_{out}$ +7.7% in $\eta_{II}$ |
| [203] | Nu | NS | NS | NA | NS | NA | NC | NA | Turbulent | Measured and Dituss | Blasius | $Al_2O_3$ | Wat | Maxwell | Einstein | 0.3 vol.% | <20 | SDBS | Exp | +28.75% in Nu |
| [204] | $h_{fi}$ and $\dot{S}_{gen}$ | NS | NS | Carnot | 4350 | NC | NC | 0 | Laminar | $h_{fi}\,D/k$ | Darcey | SWCNT | Wat | Hamilton–Crosser | NS | 0.02–0.03 vol.% | NS | NA | Theo | +15.33% in $h_{fi}$ −4.34% in $\dot{S}_{gen}$ |
| [105] | $T_{out}$ and $\dot{S}_{gen}$ | 92 | NS | Carnot | 4350 | C | NC | 0 | Turbulent | Gnielinski | Petukhov and Colebrook | $Al_2O_3$ | Wat | Maxwell and Xuan | Corcione and Brinkmann | 0–4 vol.% | 25, 50, 75, 100 | NA | Theo | $\dot{S}_{gen}$ is independent on models $T_{out}$ affected by $\phi$ but not $d_p$ |
| [205] | Nu, $\dot{S}_{gen}$ and Be | NS | NS | NS | NS | NC | NC | NS | Laminar | $k(\partial\theta/\partial Y)$ | NA | Cu | Wat | Maxwell | Pac and Cho | 0–7 vol.% | 5 | NA | Num | Nu, $\dot{S}_{gen}$ and Be increased with $\phi$ |

**Table 8.** *Cont.*

| Ref. | Performance Criterion | $\varepsilon_p$ (%) | $\alpha_p$ (%) | $\eta_{max,s}$ | $T_{sun}$ (K) | $\dot{E}_{des,\Delta p}$ | $\Delta p_{header}$ | $\Delta T_{in-air}$ | Flow Regime | Nu Model | f Model | NP Type | BF Type | k Model | μ Model | Concentration | $d_p$ (nm) | Surfactant | Research Technique | Main Achievements |
|---|---|---|---|---|---|---|---|---|---|---|---|---|---|---|---|---|---|---|---|---|
| [206] | $\eta_{II}$ and $\dot{S}_{gen}$ | NS | NS | Carnot | NS | NC | NC | NS | NS | NA | NA | Graphene | Wat | NS | NS | 0.02–0.035 vol.% | NS | NA | Theo | +21% in $\eta_{II}$ −4% in $\dot{S}_{gen}$ |
| [207] | $\eta_{II}$ | 7 | 96.2 | Carnot | NS | C | NC | −11 to 113 | Laminar and Turbulent | Li-Xuan and Rhosenow | Darcey and Blasius | Al$_2$O$_3$ | Wat | Maxwell | Batchelor | 0–3.5 vol.% | 15 | NA | Theo | +1% in $\eta_{II,opt}$ |
| [208] | $\dot{S}_{gen}$, pH, dp | 92 | NS | Carnot | 4350 | C | NC | 0 | Turbulent | Gnielinski | Petukhov | SiO$_2$ | Wat | Xuan | Brinkmann | 1 vol.% | 12 and 16 | NA | Ana | pH decreased $\dot{S}_{gen\,12nm}$ but increased $\dot{S}_{gen\,16nm}$ |
| [209] | $\eta_{II}$ | 7 | 96.2 | Carnot | NS | C | NC | −11 to 113 | Laminar and Turbulent | Li-Xuan | Darcey and Blasius | Al$_2$O$_3$ | Wat | Maiga | Maiga | 0–1 vol.% | 15 | SDBS | Exp and Theo | Max $\eta_{II,\,opt}$ = 12.53% |
| [210] | $\eta_{II}$ and Be | NS | NS | Carnot | 4350 | C | NC | NA | Laminar | NA | Darcey | SiO$_2$ | W:EG | NS | NA | 0–1 vol.% | 40 | NU | Exp | +62.7% in $\eta_{II}$, Be increased with $\phi$ |
| [211] | $\eta_I$, $\eta_{II}$ and Be | 12 | NS | Carnot | NS | C | NC | NA | Laminar | NA | Darcey | MgO | Wat | Measured | Measured | 0–2 vol.% | 40 | CTAB | Exp | +9.34% in $\eta_I$, +32.23% in $\eta_{II}$, Be approached 1.0 |

**Table 8.** *Cont.*

| Ref. | Performance Criterion | $\varepsilon_p$ (%) | $\alpha_p$ (%) | $\eta_{max,s}$ | $T_{sun}$ (K) | $E_{des,\Delta p}$ | $\Delta p_{header}$ | $\Delta T_{in-air}$ | Flow Regime | Nu Model | f Model | NP Type | BF Type | k Model | μ Model | Concentration | $d_p$ (nm) | Surfactant | Research Technique | Main Achievements |
|---|---|---|---|---|---|---|---|---|---|---|---|---|---|---|---|---|---|---|---|---|
| [212] | $\eta_I$ and $\eta_{II}$ | NS | NS | Carnot | NS | NS | NA | NA | NS | NS | NS | C Fe$_3$O$_4$ Ag | Wat | NS | NS | 5–40 ppm | 40 15 20 | TPABr | Exp | $\eta_{IandII, Fe3O4}$ > others |
| [213] | $\eta_I$, $\eta_{II}$ and Be | 12 | NS | Carnot | NS | C | NC | NA | Laminar | NA | Darcey | MgO + MWCNT CuO + MWCNT | Wat | Measured | Measured | 0–2.25vol.% | 40 + 7 42 + 7 | NS | Exp | +16.28% in $\eta_I$ +25.1% in $\eta_{II}$, Be approached 1.0 |
| [214] | $\eta_I$ | NS | 95 | NA | NS | NA | NA | 5–40 | NS | NA | NA | Graphene | Wat | Maxwell | NS | 0.025, 0.075, 0.1 wt.% | 2 | NS | Exp | +18.2% in $\eta_I$ at 0.1 wt.% |
| [215] | $\eta_I$ and $\eta_{II}$ | NS | NS | Carnot | NS | NA | NA | NS | NS | NA | NA | Al$_2$O$_3$ | Wat | NA | NA | 0.1vol.% | 20 | NS | Exp | +30.7% in $\eta_I$ +18.7% in $\eta_{II}$ |
| [216] | $\Delta p$ and $V$ | NS | NS | NA | NS | NA | C | NS | Turbulent | NA | NS | Al$_2$O$_3$, TiO$_2$ Zno | Wat | Maxwell | Einstein | 0.1vol.% | NS | NA | Modeling | Average number of risers perform better |
| [217] | $\eta_I$ | 12 | NS | NA | NS | NA | NA | 38–58 | Laminar | Heaton | NA | SiO$_2$ | Wat | Measured | Measured | <0.6vol.% | 20–30 | NS | Exp and Ana | +55.2% in $F_R U_L$ |

**Table 8.** *Cont.*

| Ref. | Performance Criterion | $\varepsilon_p$ (%) | $\alpha_p$ (%) | $\eta_{max,s}$ | $T_{sun}$ (K) | $E_{des,\Delta p}$ | $\Delta p_{header}$ | $\Delta T_{in-air}$ | Flow Regime | Nu Model | $f$ Model | NP Type | BF Type | $k$ Model | $\mu$ Model | Concentration | $d_p$ (nm) | Surfactant | Research Technique | Main Achievements |
|---|---|---|---|---|---|---|---|---|---|---|---|---|---|---|---|---|---|---|---|---|
| [218] | $\eta_I$ | 7 | 96.2 | NA | NS | NA | NA | 4–40 | NS | NA | NA | CuO | Wat | NA | NA | 0.1 vol.% | 40 | NS | Exp | +55.2% in $\eta_I$ at 4 Lit/min |
| [219] | $\eta_I$ | NS | NS | NA | NS | NA | NA | 1–6 | Laminar | Heaton | NA | $Al_2O_3$ | Wat | Maxwell | Kitano | 0.25–5 vol.% | 11 | NS | Exp and Theo | −11.7 % in $\eta_I$ |
| [220] | $\eta_I$ | NS | NS | NA | NS | NA | NA | 2–5 | NS | NA | NA | $Al_2O_3$ | Wat | Yu and Choi | Drew and Passman | 0.1–0.3vol.% | 10–15 | SDS | Exp | +21.3% in $\eta_I$ |

* NA: not applicable. NC: not considered. NS: not specified. NU: not used. Exp: experimental. Ana: analytical. Num: numerical. Theo: theoretical.

One of the important challenges in the study of nanofluid-based FPCs could be the model of exergy factor. The table shows that all the studies on exergetic performance of solar collectors focused on PLP or Carnot models. First of all, it should be noted that the Carnot and PLP models were developed for the case in which the radiation contains a blackbody absorber/emitter. Therefore, their application is limited to special conditions and is not appropriate for the collectors with spectrally selective absorbers. This is the fact that has been overlooked in the reported studies on exergy efficiency of nanofluid-based FPCs or other solar collectors. A more precise and realistic exergy factor is proposed by [125] based on the Badescu model for application in FPCs:

$$
\eta_{max,s} = \begin{cases} \text{conversion not allowed; } \frac{f_s}{\varepsilon} < a^3 \\ 1 - \frac{4}{3}a + \frac{1}{3}\frac{f_s}{\varepsilon}a^4; \frac{f_s}{\varepsilon} \in \left[a^3, 1\right] \end{cases} \tag{15}
$$

where, $a = T_a/T_s$, and $f_s$ and $\varepsilon$ stand for the geometry factor of solar radiation resource and emittance of the collector absorber, respectively. This model is valid for all sorts of solar collectors. In fact, both Carnot and PLP are a special case of this realistic model. Second, for the collectors in which $f_s < a$, the conversion of radiation energy to thermodynamic work or exergy is not allowed. Assuming the values of $T_a = 300$, $T_s = 6000$, and thus $a^3 = 1.25 \times 10^{-5}$, and $f_s = 2.176 \times 10^{-5}$, it could be shown that a collector with the emittance of $\varepsilon = 0.92$ would not be able to convert the radiation heat to work. Therefore, the exergy analysis for a such collector would not make sense. However, an enhanced absorber possessing an emittance of 0.1 would be able to generate work, since in this case, $a^3 < (f_s/\varepsilon = 2.176 \times 10^{-4})$. Hence the selection of the collector is very important to enhance the exergy efficiency and its analysis. According to this model, the ratio of the absorber plate absorptance to its emittance is very crucial as well. It is shown that only for a certain range of this parameter, the conversion of the radiation energy to thermodynamic work would be possible [221]. Therefore, the value of the absorptance of the selected collector is also of the importance to be considered in exergy analysis of the nanofluid-based FPCs. Some reported collector specifications in Table 8 for exergetic investigation do not fulfil the constraint in the mentioned model.

The exergy efficiency of a nanofluid-charged FPC can be estimated using the following equation:

$$
\eta_{exg} = \eta_{II} = \frac{\dot{m}\left[C_{p,nf}\left(T_{out} - T_{in} - T_a \ln(T_{out}/T_{in})\right) - \Delta p / \rho_{nf}\right]}{G_t A_c \eta_{max}} \tag{16}
$$

where $\Delta p$ represents the pressure drop across the collector. As seen from Table 8, the pressure drop in FPC manifolds has been overlooked in the reported studies for exergy efficiency. It can be modelled as stated by the first term in Equation (17) [222]:

$$
\Delta p = \left[f\frac{L}{D}\rho_{nf}\frac{V^2}{2}\right]_{in/out} + \left[\rho_{nf}g(L\sin\beta + h_L)\right]_{riser+fitting} \tag{17}
$$

The second term is associated with the pressure drop in riser tubes and piping fittings in which the head loss is estimated using the following expression:

$$
h_L = \frac{8\dot{m}_{riser}^2}{\rho_{nf}g\pi^2 D_i^4}\left(f\frac{L}{D_i} + \sum K_L\right) \tag{18}
$$

where $f$ is friction factor in the relevant flow regime, $L$ is the length of riser pipe, and $K_L$ denotes minor loss coefficient.

## 7. Economic Performance of Nanofluid-Based FPCs

According to past and recent studies, it is perceived that the thermal properties of nanofluid, especially thermal conductivity, can improve the efficiency of a conventional solar FPC.

However, exploiting nanofluid is associated with challenges from an economical point of view [46]. The choice of specific materials for a solar collector system is based on a trade-off analysis of cost and performance. The design and operation of a solar energy system is concerned with obtaining minimum cost of energy including all fixed and operating expenses. Having a solar collector with efficiency lower than the amount which is technologically possible may be more desirable if the cost can significantly be reduced. The objective of the economic analysis can be deemed as a determination of the least costly method for fulfilling energy demand. In other words, the collector should deliver the most BTUs for the least money. Overall, the investment cost of a nanofluid-filled solar collector is estimated as follows [223]:

$$C_s = C_f + C_a A + C_{np} \tag{19}$$

where $C_f$ is the collector area independent costs, $C_a$ is the collector area dependent costs, $A$ is the collector area, and $C_{np}$ is the costs for nanomaterial, including purchase or preparation, refilling, etc. [224]. A model for estimating the investment costs of a nanofluid-based FPC is proposed by the following [225]:

$$C_{inv} = \zeta \{ a_1 A_{abs}^{b_1} + a_2 A_{ris}^{b_2} + a_3 \forall_{ins}^{b_3} + a_4 A_{cov}^{b_4} \} + a_5 \dot{W}_p^{b_5} + C_{np} \tag{20}$$

where $\zeta, A_{abs}, A_{ris}, \forall_{ins}$, and $A_{cov}$ are collector assembly factor, absorber surface area, riser outside surface area, insulation volume, and cover surface area, respectively. $a_1$ to $a_5$ and $b_1$ to $b_5$ are constants determined based on the market price for collector components. The pumping load for nanofluid circulation, $\dot{W}_p$, is calculated by considering the pump efficiency, $\eta_p$, as follows:

$$\dot{W}_p = \frac{\dot{m}\Delta p}{\eta_p \rho_{nf}} \tag{21}$$

The required quantity of nanomaterials for the collector can be estimated by considering the number and length of riser tubes, $N_r$ and $L_r$, nanofluid concentration, $\phi$, and nanoparticle density, $\rho_{np}$, as follows:

$$m_{np} = \chi N_r L_r \left( \pi D_i^2 / 4 \right) \phi \rho_{np} \tag{22}$$

The factor $\chi$ represents the additional required volume to fill the connecting pipes and other equipment.

The purchase cost of certain nanoparticles, $C_{np}$, is calculated as follows:

$$C_{np} = m_{np} \cdot C_{nm} \tag{23}$$

where $c_{nm}$ is the unit cost of desired nanomaterials in \$/gram according to the market price.

The operating cost, $C_{op}$, is also necessary to consider in an economic evaluation of a nanofluid-based FPC [225]:

$$C_{op} = C_{elec} \cdot N_h \cdot \dot{W}_p \tag{24}$$

where $C_{elec}$ represents the unit cost of electricity per kilowatt hour (kWh), and $N_h$ stands for the number of operation hours of an FPC in a year.

It is estimated that the maintenance cost of an FPC system is 1% of the investment cost and is set to be increased by 1.0% annually [224]. On the other hand, the maintenance cost also includes the cost of changing nanofluids regularly. This is due to the challenging fact that the nanoparticles will undergo aggregation after some time, which may clog the riser tubes eventually. Therefore, a change of nanofluid is necessary around every three months (maximum stability duration reported yet [226]), which would be four times a year. Hence, the maintenance cost can be expressed as follows:

$$C_m = 0.01\, C_s + 4\, C_{np} \tag{25}$$

Now, the total annual cost is given by the following equation:

$$C = C_s + C_{op} + C_m \tag{26}$$

On the one hand, emissions and destruction saving costs, and on the other hand, the additional cost due to nanofluid application, should be considered in an economic analysis [227].

It is common to use equivalent annual cost (EAC) for evaluation of the economics of solar flat-plate collectors or similar engineering systems. To this end, the following formula can be used:

$$C_{total} = aC_s + C_{op} + C_m \tag{27}$$

where *a* is annual cost coefficient or sinking fund factor (SFF) to convert the annualized value of the investment, which can be calculated using the following equation:

$$a = \frac{i}{1 - (1+i)^{-y}} \tag{28}$$

where *i* and *y* denote the interest rate and the lifespan of an FPC system, respectively.

The technical and economic aspects of flat-plate collectors are considered simultaneously when evaluating the performance of the system, especially in the case of optimization studies. The techno-economic model as described here was utilized by [225] to optimize the performance of an $Al_2O_3$–water nanofluid-charged FPC. The particle swarm optimization (PSO) method was used to investigate the optimum values of equivalent annual cost ($C_{tot}$) in terms of \$/year and collector efficiency ($\eta_{col}$). Figure 11 shows the variation of both objective functions on a Pareto frontier basis for nanofluid mass flow rate of 0.2 kg/s and 0.3 kg/s.

As is observed, in both cases, i.e., with and without nanofluid application, collector annual cost and efficiency come into conflict with each other, implying that the improved efficiencies correspond with deteriorated annual costs and vice versa. The conflicting trend is more extensive at higher possible optimum efficiencies. In other words, the positive influence of nanoparticles on optimum values of efficiency and cost is more evident in higher efficiencies. Moreover, at both mass flow rates, the Pareto front containing optimum solutions in the case of nanofluid dominates the one in the case of water as the working fluid. Furthermore, it may be seen that the less techno-economic amelioration of a solar FPC using nanofluid will be achieved at higher mass flow rates.

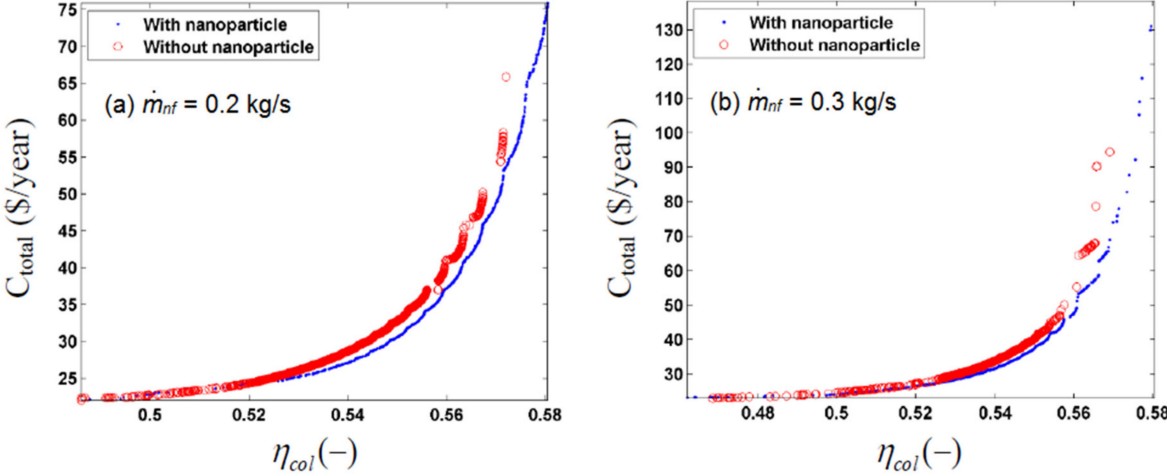

**Figure 11.** Optimum techno-economic values in the form of a Pareto frontier for an FPC with and without nanofluid at (**a**) mass flow rate of 0.2 kg/s and (**b**) mass flow rate of 0.3 kg/s.

Utilizing simultaneously nanofluid and wire coil with core rod (WCCR) inserts can influence the cost saving and energy efficiency of a plain FPC [228]. Figure 12 displays this simultaneous effect using $Al_2O_3$ nanoparticles and twenty-seven tube inserts with different ratios of coil pitch ($p$) to core rod diameter ($d$) denoted by WCCR-1 ($p/d = 1.79$), WCCR-2 ($p/d = 2.54$), and WCCR-3 ($p/d = 3.24$). Results are depicted for three volume concentrations, i.e., 0.1, 0.2, and 0.3%. As shown, the original price of an FPC in terms of USD is decreased by the application of various tube inserts, even in the case of water as working fluid. The reduction effect on collector total costs, $C_{tot}$, could be enhanced by increasing the loading of water–alumina nanofluid. Therefore, the maximum cost saving occurs at a 0.3% volume fraction, estimated to be \$88. This cost saving would be equivalent to a 26.4% enhancement in energy efficiency. This comes from the fact that the required collector area decreases significantly by improvement in the heat transfer coefficient due to the addition of nanofluid and inserts as turbulence promoters. The same behavior is observed for collector embodied energy, as indicated in Figure 13 [228].

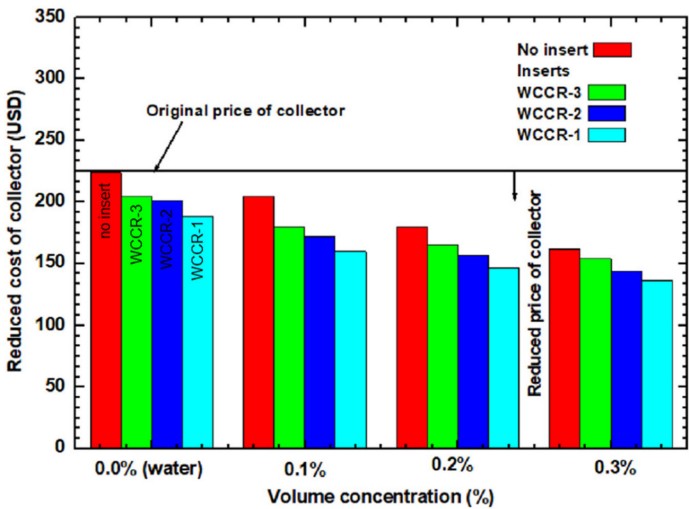

**Figure 12.** Effect of simultaneous application of nanofluid and tube inserts on the total cost of FPC.

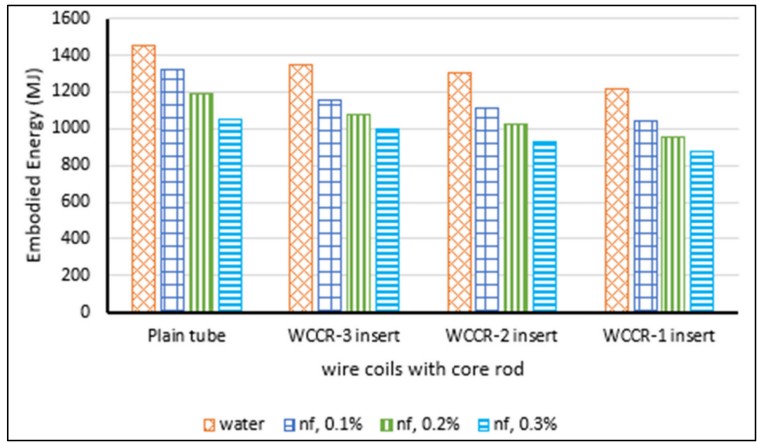

**Figure 13.** Effect of simultaneous application of nanofluid and tube inserts on FPC embodied energy.

At a concentration of 0.3%, the embodied energy is reduced by 27.17% and 39.40% utilizing nanofluid (nf) only and simultaneous application of WCCR-1 insert and nanofluid, respectively.

The main achievements from the investigations on the economic and environmental aspects of nanofluid-based FPCs are compared in Table 9.

**Table 9.** Comparative summary of performed research investigations for economic and environmental performance of nanofluid-filled FPCs. [†]

| Ref. | Performance Criterion | $\varepsilon_p$ (%) | $\alpha_p$ (%) | $\eta_{ex,max}$ | $T_{sun}$ (K) | $E_{des,\Delta p}$ | $\Delta p_{header}$ | $\Delta T_{in-air}$ | Flow Regime | Nu Model | $f$ Model | NP Type | BF Type | $k$ Model | $\mu$ Model | Concentration | $d_p$ (nm) | Surfactant | Type of Research | Main Achievements |
|---|---|---|---|---|---|---|---|---|---|---|---|---|---|---|---|---|---|---|---|---|
| [15] | $C_m$ | NA | NA | NA | NA | NA | NA | NA | Laminar and turbulent | Hausen | Darcey and Fanning | Al$_2$O$_3$, CuO, TiO$_2$ | Wat and W:EG | Corcione | Corcione | 0–0.05 vol.% | 25–100 | NA | Theo | $C_{m,min} = f(\phi_{opt})$ |
| [228] | $\eta_I$ and $c_{tot}$ | NS | NS | NA | NS | NA | NA | NA | turbulent | experimental and Gnielinski | experimental and Blasius | Al$_2$O$_3$ | Wat | Maxwell | Einstein | 0.1, 0.2, and 0.3 vol.% | NS | SDBS | Exp | +26.4% in $\eta_I$ −88 \$ in $C_{tot}$, and −39.4% in embodied energy |
| [229] | $C_{mfr}$ and CO$_2$ | 12 | 94 | Carnot | 4500 | C | NC | NA | Laminar | Choi | Darcey | Al$_2$O$_3$, CuO, TiO$_2$, SiO$_2$ | Wat | Hamilton–Crosser | Measured | 0.2 and 0.4 vol.% | 15 | NS | Exp | −220 MJ in embodied energy Payback = 2.4 yr. −170 kg in emission |
| [230] | $\eta_I$, $C_m$, $C_{mfr}$, and CO$_2$ | NS | NS | NA | NA | NA | NA | NA | NS | NA | NA | Fe | W:PG | Measured | NA | 0.5 wt.% | 40 | NS | Exp | +9% in $\eta_I$ −28.5% in annual cost −9.5% in embodied energy −37% in CO$_2$ |

**Table 9.** *Cont.*

| Ref. | Performance Criterion | $\varepsilon_p$ (%) | $\alpha_p$ (%) | $\eta_{ex,max}$ | $T_{sun}$ (K) | $E_{des,\Delta p}$ | $\Delta p_{header}$ | $\Delta T_{in\text{-}air}$ | Flow Regime | Nu Model | $f$ Model | NP Type | BF Type | $k$ Model | $\mu$ Model | Concentration | $d_p$ (nm) | Surfactant | Type of Research | Main Achievements |
|---|---|---|---|---|---|---|---|---|---|---|---|---|---|---|---|---|---|---|---|---|
| [225] | $\eta_I$ and $C$ | 92 | NS | NA | NS | NA | NC | 0 | Turbulent | Vajjha | Fanning | $Al_2O_3$ | Wat | Calvin–Petersona | Wang | 0–0.1 vol.% | NS | NA | Theo | +2% in $\eta_I$ −3.5% in $C$ |
| [198] | $C$ and $C_{mfr}$ | 95 | NS | NA | NS | NA | NA | NA | NS | NA | NA | $CeO_2$ | Wat | Xuan | Corcione | 0.01 vol.% | 25 | NS | Exp and Theo | −11.5% in total cost −28.9% in embodied energy |
| [231] | $CO_2$ | NS | 95 | NA | NS | NA | NA | NS | NS | NA | NA | $Al_2O_3$ | Wat | NS | NA | 1.5 wt.% | NS | NA | Theo | −31% in kg-$CO_2$/kWh |
| [232] | $\eta_I$, $CO_2$ and $SO_2$ | NS | 95 | NA | NS | NA | NA | NA | NS | NA | NA | $Al_2O_3$ | Wat | NS | NA | 0.5, 1, and 1.5 vol.% | 20, 50, and 100 | NS | Exp | +14.8% in $\eta_I$ −190 kg, −557 kg, and −2.03 kg in Coal, $CO_2$, and $SO_2$ |

† NA: not applicable. NC: not considered. NS: not specified. Exp: experimental. Ana: analytical. Num: numerical. Theo: theoretical.

It is inferred from the table that economic studies on nanofluid-based FPCs are mostly limited to the estimation of embodied energy and emissions reduction. However, the payback period and other economic criteria of nanofluid applications in nanofluid-filled FPCs have not yet been covered by the reported investigations. Reported techno-economic studies also have not dealt with this matter thoroughly.

It is worth mentioning that the concepts of efficiency, economic and environmental, are interrelated. The exergy efficiency is a tool for assessment of environmental impacts of a process from an economic standpoint as well [233]. This originates from the fact that the exergy concept explains the change of thermodynamic properties compared to a standard environment as reference. Generally, the improvement of inefficiencies in a thermal system such as solar FPC can result in lower exergy destruction, a more economic system, and lower environmental impacts. That is why in recent years, exergoeconomic (EXEC), exergoenvironmental (EXEN), and exergoenviroeconomic (EXENEC) concepts have been developed for comprehensive analysis of a thermal system. When performing an EXENEC study, three main components, viz. exergy, economics, and environmental aspects are considered.

## 8. Summary and Conclusions

A detailed literature review on nanofluid application in solar FPCs is presented above. Different nanomaterials and their thermophysical properties, the existing classical and empirical models for evaluation of thermophysical properties and heat transfer coefficient, various used base liquids and nanofluids at different sizes and concentrations, and the main achievements of almost all investigations for nanofluid-based FPCs are included in this review. Conclusions drawn from the review and comparison are summarized below.

- Some inconsistencies between the results of different studies have been observed, especially for pressure drop change with nanofluid working flow. Moreover, the linear friction losses in manifolds of FPCs are also important when estimating the performance of nanofluid-based FPCs and must be considered. A literature survey showed that the pressure drop in headers is overlooked in almost all studies on nanofluid-filled FPCs and should be considered.

- Most reported investigations analyzed the thermal energy enhancement for the zero-loss point at which the ambient temperature and collector inlet temperature are equal. This is very different from the nominal working conditions of a real solar FPC. In fact, it shows the conditions at which there is no thermal absorption and only the optical performance of the collector would be regarded. This is why the reported enhancements for thermal efficiency of nanofluid-based FPCs are so high. It is required in future investigations to evaluate the overall efficiency of the collector operating with nanofluid as the absorbing agent.

- All investigations carried out thus far for exergy balance of conventional and nanofluid-based FPCs have used the Carnot and PLP models for incident radiation exergy. It is realized, based on many research studies, that the apparent temperature of the sun is taken as 4500 K, while the valid quantity is 5770 K, or it is said to be 6000 K. It is less known that these models were developed for the case where the ambient consists of radiation emitted by a blackbody collector. Therefore, their usage is rigorously restricted to the case where the conversion of radiation energy into work is performed by using blackbody collectors. This fact may confront the exergy evaluation of FPCs, a conceptual challenge, since spectrally selective (not blackbody) absorbers are used in FPCs. Therefore, a more elaborated and precise exergy factor model should be adopted.

- Considering the mean temperature of 6000 K and 300 K for the sun and ambient air, respectively, the maximum exergy efficiency of an FPC cannot exceed 5%. Therefore, the exergy efficiencies that have been reported by some investigations need to be revisited.

- The combined application of selective absorbers and nanofluids for improving the power generation by FPCs has not been reported in the current literature. The key role of the ratio of the

absorber plate's emittance to its absorptance must be given attention, since only certain values are allowed by the second-law requirements.

- The absorber plate of a solar FPC is at the same time an emitter of the received radiation. The exergy loss due to the radiation emission by the absorber plate is also of importance to be discussed and considered.

- Regarding the fact that using nanofluids instead of water as HTF in conventional FPCs will affect all thermophysical properties including conductivity from one side and viscosity and density from the other side, a comprehensive techno-economic optimization is very crucial. This optimization should cover thermal efficiency, exergy efficiency, and economic performance of an FPC, which is going to be operated with a nanofluid working agent. Such a techno-economic optimization, called multi-objective optimization, for the performance of a nanofluid-charged FPC has not been reported yet by the current literature. Furthermore, very limited numbers of practical investigations with a standard and commercial SWH based on FPC have been reported.

- The existing correlations for evaluating the behavior of different nanofluids are not general and most of them are applicable for special conditions in terms of concentration and size of NPs. Therefore, still there exists a substantial need for development of a generalized model for the thermophysical properties of different common nanofluids. The stability of the nanofluids, especially at temperatures higher than 60 °C, remains a problem that has not been dealt with in carried-out investigations. It is not very easy to have stable nanofluids at volume concentrations higher than 1% in practice. In this regard, it should be noted that using surfactants will cause the conductivity of nanofluid to decrease. The stability problem would be critical for dense nanofluids, e.g., Cu and CuO nanoparticles. Therefore, selecting concentration ranges beyond 1.0% should be made with enough attention to the consequent issues.

- With regard to the importance of sustainable development, it is essential to use materials that are less chemically contained. In other words, it would be preferred to exploit bio-based materials to advance the renewable cycle as much as possible. However, it is inferred from the literature that ethylene glycol (EG), propylene glycol (PG), and polyethylene glycol (PEG) have been frequently used as base liquid mixtures with water. All of them are chemically-derived or even, like EG for instance, may have toxic effects on the environment. Any bio liquid for a mixture base fluid has been used for FPCs, while bio-glycol (BG), which is a renewable-derived product, possesses the capability to be investigated for this purpose. A comparative study on the effects of these products seems to be essential in future investigation in the field of nanofluid-based FPCs.

- In most of investigations, the specification of the FPC is not given. First, this can be a problem for the other researchers to repeat the study. Second, the important data like absorptivity and emissivity of the absorber plate should be known to assess its capability of work extraction. This would be very important when the subject matter is exergy analysis and work extraction of a nanofluid-based FPC. The assessment would be possible by using the existing equations and considering the ratio of these two properties.

- The models for estimating different parameters, specification of the selected nanoparticles, and flow regime under which the analysis has performed are not disclosed in some research works. The most important parameter for an FPC is the mass flux of working fluid, which is the amount of mass flow per unit surface area of the collector. The mass flux is usually recommended to be in a certain range with regard to the design features by the manufacture. The change in collector flow rate suggested by some research studies should be analyzed considering this range. Different mass fluxes from the design or nominal point might change the collector performance, e.g., the rate of heat transfer and pipe corrosion speed.

- Economic and environmental aspects that previously been studied have not covered the details of the application of nanofluids in FPCs. Adoption of nanofluid application in FPCs should be subjected to financial feasibility to evaluate the success of the investment. Not only the material costs should be considered, as the operating costs are vital parameters, as well in the feasibility

analysis. For instance, since nanofluids are stable for only for a few months, they need to be changed at least three or four times per year. The running and operational costs should be considered in economic analysis, along with purchase or preparation costs, and the relevant payback period should be analyzed.

- Sustainable accomplishment can be enhanced or lessened depending on factor(s) of consideration. It may be evaluated by many factors on adoptability of the nano tech in flat-plate solar systems. These factors may be cost effectiveness, reduction in $CO_2$ emission, and reduction of environmental harms.
- Adopting bio blended base fluid as thermal working fluid in FPCs is environmentally-friendly compared to the chemical product's base fluids.
- The developed cost analysis method will provide a tool for the industry and investors on solar energy via FPCs to analyze and judge the adaptability of the nano-enhanced thermal fluids for FPCs.
- Lastly, wider adoption of FPCs for domestic and industrial applications with higher efficiency and/or lower cost in the long term will increase the utilization of solar energy for the heating process. As a clean source, it will impact the reduction of $CO_2$.

**Author Contributions:** Conceptualization, S.R.S. and H.H.A.-K.; methodology, S.R.S., H.H.A.-K. and K.V.S.; investigation, S.R.S.; resources, H.H.A.-K.; data curation, S.R.S. and M.G.; writing—original draft preparation, S.R.S.; writing—review and editing, H.H.A.-K. and K.V.S.; supervision, H.H.A.-K. and K.V.S.; project administration, H.H.A.-K.; funding acquisition, H.H.A.-K. All authors have read and agreed to the published version of the manuscript.

**Funding:** This research was funded by national fund grant YUTP-FRG, grant number 015LC0-026 from PETRONAS, Malaysia.

**Acknowledgments:** The authors acknowledge PETRONAS and Universiti Teknologi PETRONAS (UTP) for supporting the work under research grant YUTP-FRG with CS: 015LC0-026. The first author expresses his appreciation to University Teknologi PETRONAS for the financial and technical support to conduct his PhD research in the Solar Thermal Advanced Research Centre (STARC).

**Conflicts of Interest:** The authors declare no conflict of interest.

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
