# Peer review of "State of the Art of Techno-Economics of Nanofluid-Laden Flat-Plate Solar Collectors for Sustainable Accomplishment"

_sustainability, doi:10.3390/su12219119_

Round 1

Reviewer 1 Report

Dear authors, I would like to note the following suggestions and comments for your paper

1. The authors do not disclose the concept of "Sustainable Accomplishment". This phrase appears in the title of the paper. However, it is not clear how "Nanofluids-Laden Solar Flat Plate Collectors" ensure the "Sustainable Accomplishment" of energy systems.

2. Despite the fact that the article should include a technical and economic analysis, based on the title of the article, the economic aspects of the study are very poorly represented.

3. The article does not include the "Materials and Methods" section.

4. I would also like to see the "Discussion" section.

In General, the authors have done a great job.

Reviewer 2 Report

Interesting review article.

Contains both historical and current literature sources. The authors themselves also have achievements in the described field.

There is a certain lack of description of the application of the described devices, but this is not necessary in the review article and in my opinion it does not require any special supplement.

Author Response

Responds to Reviewer 2

Reviewer:

Interesting review article.

Contains both historical and current literature sources. The authors themselves also have achievements in the described field.

There is a certain lack of description of the application of the described devices, but this is not necessary for the review article and in my opinion, it does not require any special supplement.

Authors respond:

Thank you for reviewing the manuscript. We appreciate your understanding of the nature of the review paper. We agree with you that there no need to add a description of the application of the devices. 

Reviewer 3 Report

An interesting review manuscript has been submitted by the authors, well done and with good scientific soundness, founding the start of art of technology and economic of Nano fluids on solar flat plate collection.

Author Response

Responds to Reviewer 3

An interesting review manuscript has been submitted by the authors, well done and with good scientific soundness, founding the start of the art of technology and economic of Nanofluids on solar flat plate collection.

Author responds:

Thank you for the time and effort to review our long article and the support for publishing the article. Appreciate your support.